# Targeted small molecule inhibitors blocking the cytolytic effects of pneumolysin and homologous toxins

Umer Bin Abdul Aziz[1,8], Ali Saoud[1,8], Marcel Bermudez[1,2], Maren Mieth[3], Amira Atef[1,4], Thomas Rudolf[1], Christoph Arkona[1], Timo Trenkner[5], Christoph Böttcher[6], Kai Ludwig[6], Angelique Hoelzemer[5,7], Andreas C. Hocke[3], Gerhard Wolber[1] & Jörg Rademann[1]✉

Pneumolysin (PLY) is a cholesterol-dependent cytolysin (CDC) from *Streptococcus pneumoniae*, the main cause for bacterial pneumonia. Liberation of PLY during infection leads to compromised immune system and cytolytic cell death. Here, we report discovery, development, and validation of targeted small molecule inhibitors of PLY (pore-blockers, PB). **PB-1** is a virtual screening hit inhibiting PLY-mediated hemolysis. Structural optimization provides **PB-2** with improved efficacy. Cryo-electron tomography reveals that **PB-2** blocks PLY-binding to cholesterol-containing membranes and subsequent pore formation. Scaffold-hopping delivers **PB-3** with superior chemical stability and solubility. **PB-3**, formed in a protein-templated reaction, binds to Cys428 adjacent to the cholesterol recognition domain of PLY with a $K_D$ of 256 nM and a residence time of 2000 s. It acts as anti-virulence factor preventing human lung epithelial cells from PLY-mediated cytolysis and cell death during infection with *Streptococcus pneumoniae* and is active against the homologous Cys-containing CDC perfringolysin (PFO) as well.

Pneumolysin (PLY) is a pore-forming toxin belonging to the family of cholesterol-dependent cytolysins (CDC)[1,2]. The 53 kDa protein is a major virulence factor of *Streptococcus pneumoniae* (*S.pn.*), which is the leading bacterial cause of community-acquired pneumonia, meningitis, and otitis media[3,4]. Despite the broad availability of antibiotics against *S.pn.*, every year around two million patients die from pneumonia worldwide, primarily infants and elderly people[5,6]. The virulence factor PLY is considered widely as a main trigger of morbidity and mortality in invasive *S.pn*[7]. The protein is expressed within nearly all known serotypes and released from the bacteria as soon as the cells

are ruptured due to auto- or antibiotic-mediated lysis[4,8,9]. As a consequence, the toxin can exert active function inside the host, even in the case the antibiotics successfully killed the bacteria[5]. Therefore, there is a crucial medical need for an anti-virulence strategy able to neutralize PLY[7].

PLY comprises 471 amino acids organized in four domains[10]. C-terminal domain 4 is of the most critical value because it contains motifs like cholesterol recognition motif (CRM) and undecapeptide (UDP), which are responsible for cholesterol binding and the anchorage of PLY within mammalian cell membranes[11]. Once bound to the

[1]Institute of Pharmacy, Freie Universität Berlin, Königin-Luise-Str. 2+4, 14195 Berlin, Germany. [2]Institute for Pharmaceutical and Medicinal Chemistry, University of Münster, Corrensstr. 48, 48149 Münster, Germany. [3]Department of Infectious Diseases, Respiratory Medicine, and Critical Care, Charité - Universitätsmedizin Berlin, Corporate Member of Freie Universität Berlin and Humboldt-Universität zu Berlin, Berlin, Germany. [4]Department of Medicinal Chemistry, Faculty of Pharmacy, Assuit University, Assiut 71526, Egypt. [5]Leibniz Institute of Virology, Hamburg 20251, Germany. [6]Institute of Chemistry and Biochemistry, Research Center of Electron Microscopy (FZEM), Freie Universität Berlin, Fabeckstraße 36A, 14195 Berlin, Germany. [7]First Department of Medicine, University Medical Center Hamburg-Eppendorf (UKE), 20251 Hamburg, Germany. [8]These authors contributed equally: Umer Bin Abdul Aziz, Ali Saoud. ✉e-mail: joerg.rademann@fu-berlin.de

cell membrane, monomers of PLY oligomerize via the non-contiguous N-terminal domains 1–3 to form a pre-pore complex that subsequently undergoes a sequence of structural transitions to form a membrane-infiltrating β-barrel pore[12]. Pore formation results subsequently in cell death by cytosolic and mitochondrial calcium overload[13]. A typical pore is assembled from up to 50 PLY monomers and has an average diameter of 350 Å[12]. Pathogenesis associated with PLY, however, is not limited to cell death. PLY significantly enhances bacterial colonization throughout the host[14] and activates the immune system by triggering the inflammasome[15,16] and may also contribute to organ damage and septic shock[17].

Despite the medical need currently no clinically admitted drugs are available for the targeted treatment or prevention of PLY-mediated pathogenesis[1,18]. Unspecific phytochemicals with reported polypharmacology[5], including flavonoids such as quercetin or epigallocatechin[19–22], plant steroids[11,23], and other natural products[24–26] have been found to interfere with PLY activity, however, these molecules have been reported active toward numerous proteins and against many diseases without adequate biological or clinical evidence[27]. In addition, lipoproteins[28] and liposomes loaded with cholesterol or sitosterol[29] have been shown to scavenge PLY. The therapeutic value of such lipid-based approaches to remove PLY is, however, questionable as interference with lipid homeostasis is linked to additional health issues including atherosclerosis, cardiac infarction, and cerebral ischemia. In summary, no drug-like and directly acting, targeted PLY inhibitors have been reported so far.

In this study, we adopt a combination of in silico and in vitro methods to discover, validate, and optimize the targeted inhibitors of PLY. We report molecules able to block hemolysis exerted by PLY and suppress cytotoxicity of PLY toward human lung cells. Using cryo-TEM tomography we show that the inhibitors efficiently inhibit pore formation by this bacterial toxin. We validate the mode of action, binding site, kinetics, and specificity of inhibitors by applying bio-layer interferometry and protein mass spectrometry with wild-type and mutant PLY as well as with other proteins. Finally, we demonstrate the physiological efficacy of the best inhibitor by using an in vitro bacterial infection model in human lung epithelial cells.

## Results

### Structure-based screening for PLY inhibitors

In order to apply a structure-based screening approach for identifying inhibitors of PLY, crystallographic data of PLY monomers (PDB entry 4ZGH)[30] were fitted into a cryo-EM map of the pre-pore (PDB entry 2BK2)[31] (Fig. 1a). This provided a structural model of a PLY dimer (Supplementary Fig. 1a), which led to the identification of two potential binding sites for small organic molecules, one pocket in the cholesterol binding domain and another one in the oligomerization domain (Fig. 1b). The second pocket was located adjacent to the dimer interface in close proximity to a salt bridge formed by E151 to K288 of the adjacent monomer (Supplementary Fig. 2a, b). The PLY-mutant E151Q validated the functional relevance of this dimerization hotspot (Supplementary Fig. 2c). Moreover, the region of the proposed binding site was reported to be crucial for large-scale rearrangements upon pre-pore-to-pore formation[32]. Therefore, about 555,000 molecules were virtually screened for this binding pocket using pharmacophore-based docking, an approach widely used for the discovery of drug-like molecules[33]. The hit list was filtered by physicochemical properties and the hit molecules were re-scored by geometric fit to a 3D pharmacophore using LigandScout's geometric scoring function[33]. Finally, 10 molecules were selected after clustering for chemical diversity and visual inspection (Fig. 1c, Supplementary Table 1, Supplementary Fig. 3).

### PLY inhibition prevents hemoglobin release from erythrocytes

For testing of the virtual hit structures, recombinant PLY was expressed with a 10-fold N-terminal His-tag, and its native cytolytic activity

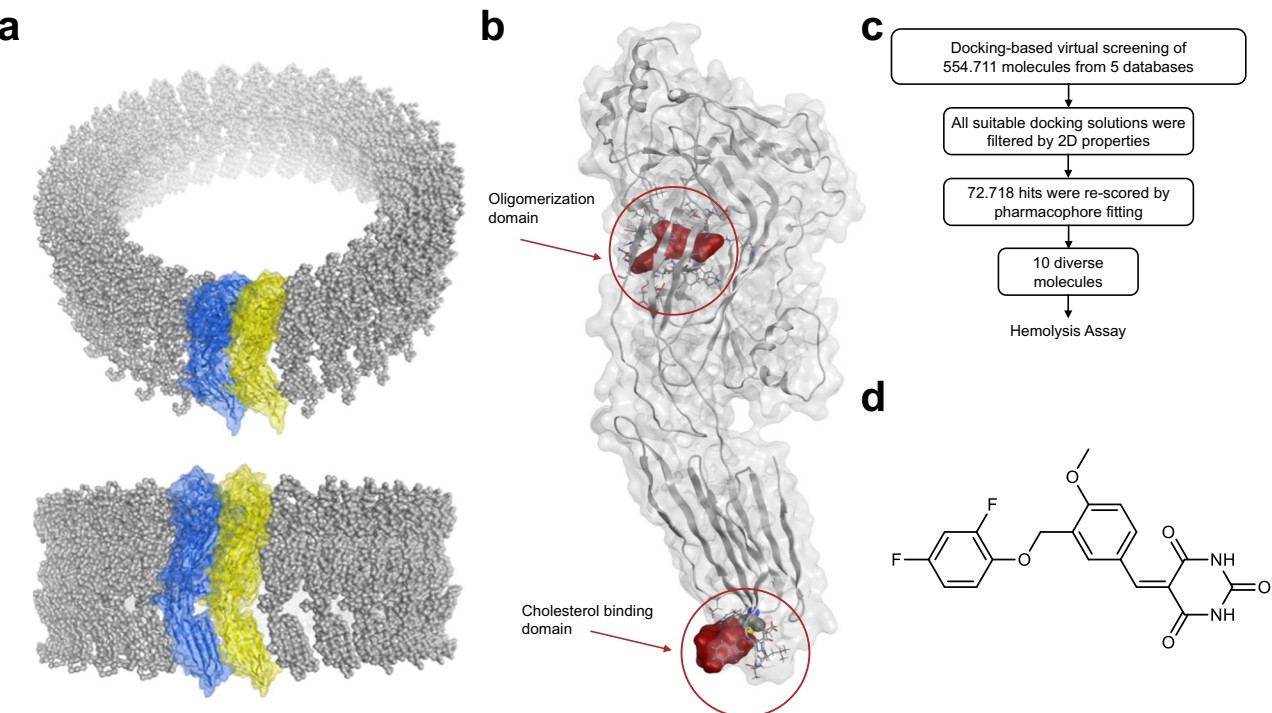

**Fig. 1 | Structure-based virtual screening for the identification of PLY inhibitors. a** A structural model of a PLY dimer (blue and yellow are two neighboring monomers) was built by fitting crystallographic data (PDB entry: 4ZGH)[30] into a cryo-EM map (PDB entry: 2BK2)[31]. **b** Identified potential binding sites shown as surface in red in the oligomerization and in the cholesterol binding domain (red circles). **c** Workflow of virtual screening resulting in 10 diverse hit molecules for testing in a hemolysis assay. **d** Molecular structure of **PB-1**.

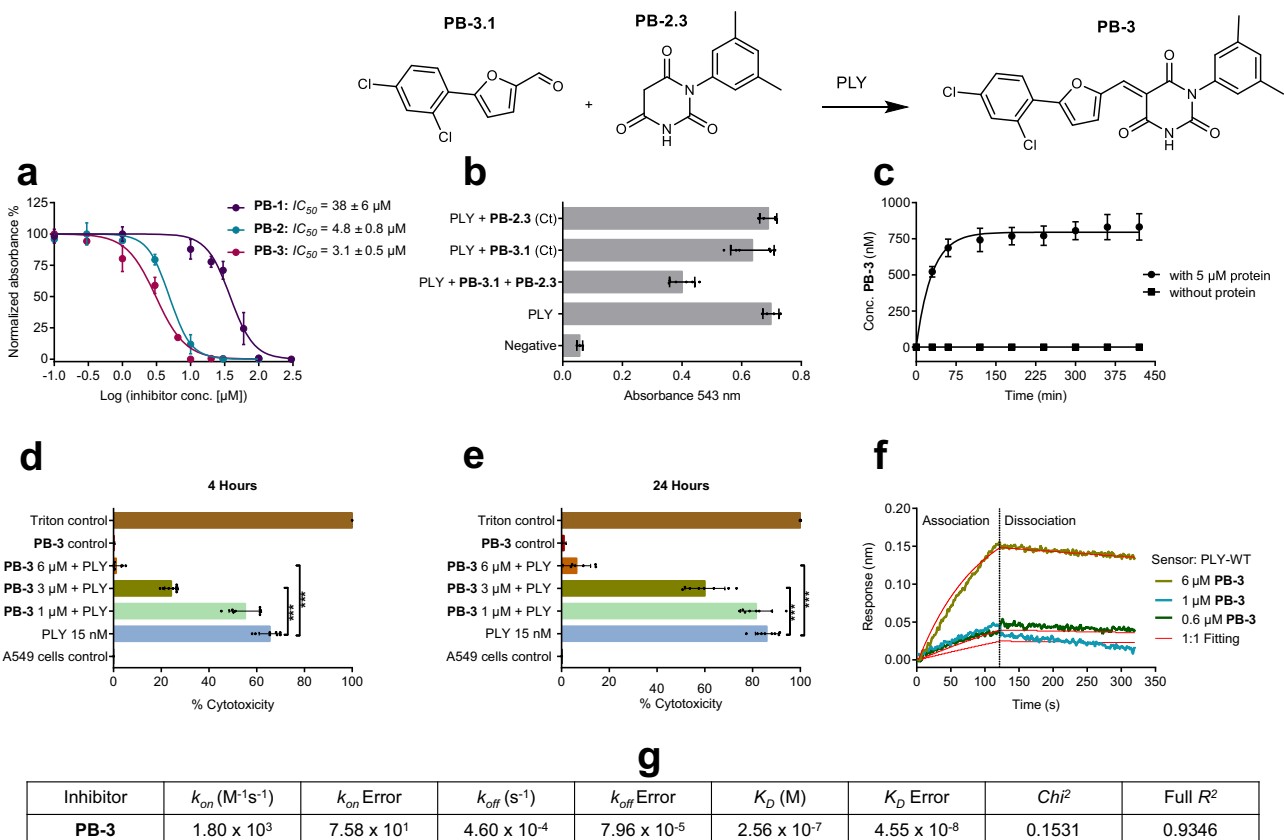

**Fig. 2 | The evaluation of activity and binding kinetics of PB-3. a** $IC_{50}$ values of **PB-1**, **PB-2**, and **PB-3** (distinct samples $n = 3$, 95% CI for $IC_{50}$ and error bars represent ± S.D.) corresponding to the ability of **PB-inhibitors** to block the PLY-induced hemoglobin release in sheep erythrocytes. **b** PLY was inhibited by **PB-3**, formed via protein-templated fragment ligation in the hemolysis assay, while **PB-3.1** and **PB-2.3** alone were no inhibitors (distinct samples $n = 3$, error bars represent ± S.D.). **c** Quantification of **PB-3** formed by protein-templated ligation using HPLC-QTOF-MS. Concentrations were fitted to the one-phase saturation equation $c(t) = c_0 + (c_{max} - c_0) * (1 - exp(-K*t))$ used to obtain $c_{max}$ (795 nM) and $t_{1/2}$ (~20.5 min) (distinct samples $n = 3$, error bars represent ± S.D.). **d**, **e** PB-3 prevents human alveolar epithelial cells from PLY-associated impairment. Samples include controls

(positive and negative), PLY alone, and PLY with **PB-3** (1, 3 and 6 μM). Figures show LDH release (which is quantified as percent-cytotoxicity) after 4 h and 24 h. Mann Whitney test (two-tailed) is used for statistical evaluation (distinct samples $n = 8$, ***$p = 0.0009$ for **d** distinct samples $n = 8$ ***$p = 0.0002$ for (**e**), and error bars represent ± S.D.). **f** Binding kinetics of **PB-3** and PLY complex measured in the BLI assay. PLY-C428A is used as reference protein and buffer is reserved as inhibitor reference. Binding is observed over 0.6, 1, and 6 μM of **PB-3** (data shown are representing one of $n = 3$ distinct samples). The reference-subtracted data are presented in the graph, which display association and dissociation steps of wild-type PLY and **PB-3**. **g** The one-to-one kinetic fitting model was used to calculate the on- and off-rates ($k_{on}$, $k_{off}$) of binding and the binding affinity ($K_D$) of **PB-3** to PLY.

was tested with sheep erythrocytes without and with the addition of human serum. The membranes of sheep erythrocytes are well-suited as targets of PLY as they contain an equally high amount of cholesterol (ca. 23%) as human cells[34,35]. The resulting $LD_{50}$ values were 0.5 nM without serum and 2.6 nM with serum, respectively (Supplementary Fig. 4). The observed increase of the $LD_{50}$ of PLY after the addition of human serum can be attributed to the abundance of cholesterol in the serum which reduces the activity of PLY[27]. Based on that, and to maintain an optimum PLY hemolytic activity during the hemolysis assay, the primary screening of PLY inhibitors was performed utilizing isolated sheep erythrocytes without serum. The inhibitory potential of 10 virtual hits (VH) was investigated at three different concentrations (250 μM, 500 μM, and 1 mM) against PLY (Supplementary Table 1, Supplementary Fig. 3). One molecule, a substituted 5-benzylidene-pyrimidine-2,4,6-trione, was actively hindering hemoglobin release at all three concentrations without precipitation and thus was named pore-blocker-1 (**PB-1**) (Fig. 1d). The $IC_{50}$ of **PB-1** toward PLY was determined in the same assay and was 38 μM (Fig. 2a, Table 1). Dynamic pharmacophores[33] were used to postulate relevant interactions of **PB-1** in the proposed binding site (Supplementary Fig. 1c, d). **PB-1** was synthesized through a Knoevenagel condensation of 3-(2,4-difluor-ophenoxymethyl)-4-methoxy-benzaldehyde with pyrimidine-2,4,6-

trione (barbituric acid). Six structural analogs of **PB-1** were selected with substructure variations in order to scrutinize the structure-activity profile of **PB-1**, resulting in the discovery of **PB-2** with an improved $IC_{50}$ value of 4.8 μM (Table 1, Fig. 2a, Supplementary Fig. 5). N-substitution of the pyrimidine ring desymmetrized the double bond and accordingly **PB-2** was obtained as a 1:1 equilibrium of the E/Z isomers which were interconverted rapidly and thus could not be separated by chromatographic methods. The observed isomerism of **PB-2** can be rationalized by a plausible hydration/dehydration mechanism (Supplementary Fig. 6). Fifteen derivatives of **PB-2** were designed and synthesized to understand the SAR of the inhibitor and to optimize it for advanced cellular applications (Table 1, Supplementary Fig. 7). **PB-2.1**, comprising a **PB-2** derivative with reduced olefinic double bond, was inactive, suggesting the functional significance of this bond for PLY inhibition. The precursor fragments of **PB-2**, aromatic aldehyde **PB-2.2**, the N-substituted barbituric acid **PB-2.3**, and the truncated precursor **PB-2.4** were inactive toward PLY in the hemolysis assay with 10 min incubation. Removal of the 3,5-dimethyl-phenyl substituent in the N1-position of **PB-2** resulting in **PB-2.5** increased the $IC_{50}$ value toward PLY more than 2-fold to 13 μM (Table 1). Further modifications of the left and middle benzene rings either reduced or abolished PLY inhibition in **PB-2.6-2.11** (Supplementary

**Table 1 | Inhibition of hemolysis exerted through wild-type PLY by pore-blocking agents PB-1 to PB-3 and derivatives (distinct samples $n \geq 3$)**

| Compound | Structure | $IC_{50}$ [μM][a] (95% CI) |
|---|---|---|
| PB-1 | | 38 ± 6 |
| PB-2 | | 4.8 ± 0.8 |
| PB-2.1 | | Inactive |
| PB-2.2 | | Inactive[b] |
| PB-2.3 | | Inactive |
| PB-2.4 | | Inactive |
| PB-2.5 | | 13 ± 1.7 |
| PB-2.12 | | 51 ± 1.6 |

**Table 1 (continued) | Inhibition of hemolysis exerted through wild-type PLY by pore-blocking agents PB-1 to PB-3 and derivatives (distinct samples n ≥ 3)**

| Compound | Structure | $IC_{50}$ [μM][a] (95% CI) |
|---|---|---|
| PB-3 | | 3.1 ± 0.5 (0.22 ± 0.09)[c] |

[a]Incubation with PLY for 10 min, for details see "Methods" section.
[b]No activity under standard conditions; prolonged incubation for 1.5 h yielded an $IC_{50}$ value of 18 μM.
[c]With 2.5% human serum. For extended SAR with 46 further derivatives, see Supplementary Figs. 7, 8, and 14.

Fig. 7). Replacement of the pyrimidine ring at the right end of **PB-2** led to reduced activities in compounds **PB-2.12-2.14** as well. For example, the condensation product obtained from aldehyde **PB-2.2** and malonic acid diamide, **PB-2.12**, displayed a residual activity with an $IC_{50}$ value of 51 μM (Table 1). These results suggested that the reactivity of the olefinic double bond in both **PB-1** and **PB-2** induced by the electron-withdrawing pyrimidine ring was essential for PLY inhibition. At the same time, a gradual reduction of potency of inhibitors **PB-1** and **2** was observed in phosphate buffer saline (PBS) and in cell culture medium, presumably via hydration followed by a retro-aldol reaction, while the molecules were entirely stable in DMSO. Systematic variation of the substituents of the reactive double bond led to the discovery of the more potent and stable molecule **PB-3**. Replacement of the substituted benzylidene residue by 5-(2,4-dichlorophenyl)-furfuryl-2-methylene in **PB-3** increased the chemical stability of the PLY inhibitor considerably with less than 5% hydrolysis after 6 h in PBS (pH 7.4) at 37 °C (see Supplementary Fig. 2b–d) and improved potency to an $IC_{50}$ of 3.1 μM (Fig. 2a). **PB-3** was the most active PLY inhibitor of 13 structural derivatives, **PB-3.2-PB-3.13** (Supplementary Fig. 8). Moreover, **PB-3** showed an enhanced activity with an $IC_{50}$ value of 0.22 μM against PLY in presence of 2.5% human serum in the hemolysis assay (Supplementary Fig. 4).

## PLY-inhibition protects human lung cells

As PLY release is associated with fatal cellular and organ damage, particularly in pneumococcal pneumonia[5,36], inhibitors **PB-1, 2,** and **3** were verified by preventing the PLY-induced cellular damage in human alveolar epithelial cells (A549 cells). Confluent cells were incubated with PLY alone, with PLY and multiple concentrations of inhibitors, and with inhibitor alone. After 4 h and 24 h, respectively, supernatant of each experiment was aspirated for analysis. Cellular lysis induced by PLY was quantified using the lactate dehydrogenase (LDH) assay because this assay provides a direct indication of cell membrane rupture due to pore formation leading to LDH release, which can subsequently be quantified. **PB-1, PB-2** (Supplementary Fig. 9), and **PB-3** (Fig. 2c, d) exhibited significant concentration-dependent inhibition of PLY and prevented more than 95% of cellular damage, for example at 6 μM for **PB-3**.

## Kinetic validation of inhibitor binding to PLY

Bio-layer interferometry (BLI) was applied to validate binding of **PB-2** (Supplementary Fig. 10a, b) and **PB-3** to PLY protein and to determine the binding kinetics. BLI is a surface-based protein-binding assay, which uses the recorded shifts in the interference pattern of white light

in order to measure the adsorption and desorption of ligands to a sensor surface loaded with protein[37]. Here, His-tagged wild-type PLY and a non-binding PLY-C428A mutant as protein reference (see below) were loaded onto a Ni-NTA-coated sensor surface and the protein-loaded biosensors were dipped in buffer solutions with three different concentrations (0.6, 1 and 6 μM) of **PB-3** prepared from a **PB-3** stock solution in DMSO. Buffer including equivalent amounts of DMSO (up to 5%) were employed as reference samples without inhibitor. Association of **PB-3** to wild-type PLY and dissociation from PLY were recorded over time when the sensor was dipped in the inhibitor solution and buffer solution without inhibitor. Protein and inhibitor reference data was subtracted and the binding isotherm was plotted with time on x-axis and response (nm) on y-axis (Fig. 2f). The 1:1 binding model was applied and kinetics were calculated by using ForteBio data analysis9 software, revealing that **PB-3** bound with an affinity or $K_D$ value of 256 nM, to wild-type PLY (Fig. 2g). The high binding affinity corresponded to a high on-rate (1800 M$^{-1}$s$^{-1}$) and small off-rate (0.00046 s$^{-1}$), furnishing a residence time of 2174 s.

## Inhibition of PLY pore formation investigated by cryo-TEM

Cryo-transmission electron microscopy (cryo-TEM) has been established as a method to study membrane integration of and pore formation by cholesterol-dependent cytolysins[12,31]. Liposomes with a molar ratio of 70% di-oleyl phosphatidyl choline (DOPC) and 30% cholesterol were prepared by extrusion and characterized by dynamic light scattering (DLS) (Supplementary Fig. 11). Recombinant wild-type PLY was incubated with liposomes, with and without inhibitor **PB-2**, and examined by cryo-TEM and cryo-electron tomography (cryo-ET). In the case of PLY added to liposomes, observations were in accordance with earlier reports[38], integration of protein into the liposomal membranes was noticed, and simultaneously events of pre-pores and pores formation were observed. The data of PLY-liposomes is presented in Fig. 3a and Supplementary Video 1 (stereogram in Supplementary Fig. 12j). Moreover, protein pores formed by oligomerization of PLY were isolated by treating the PLY-liposome mixture with Cymal-6 followed by Amphipol-35 and protein pores were also inspected by cryo-ET (Fig. 3b, Supplementary Video 2)[12]. In contrast, in the presence of inhibitor **PB-2**, naked liposomes were recognized without any membrane-integrated proteins, and due to the lacking protein-protein interactions, these liposomes were more homogenously distributed in comparison to samples without inhibitor (Fig. 3c, Supplementary Video 3). Additionally, PLY blocked by **PB-2**, was identified and highlighted as free suspended particles in the same sample volume (Fig. 3c).

## Binding site and mode of action of PLY inhibitors

To scrutinize the binding site of **PB**-inhibitors, mutational analysis of the two potential binding pockets in the oligomerization and cholesterol binding domain shown was conducted (Fig. 1). Six mutants of PLY were constructed involving amino acids in the oligomerization domain, three in the proposed binding pocket (PLY-T55A, PLY-S61A, and PLY-N85L) and three adjacent to it (PLY-E151Q, PLY-E151K, and PLY-D205R) (Supplementary Fig. 13). The three proteins mutated in the binding pocket (PLY-T55A -N85L and -S61A) exhibited hemolytic activities up to 3 times more potent than wild-type PLY and were blocked by **PB-1, 2**, and **3** with comparable or reduced $IC_{50}$ values, although the mutations abolished H-bonds predicted to participate in the binding of the **PB**-molecules (Fig. 4a, Supplementary Fig. 12d–f, h, i). The three proteins mutated outside of the binding pocket exhibited no hemolytic activity (Supplementary Fig. 12a, b), presumably as the mutated proteins were not able to oligomerize anymore, and consequently no inhibitory effect of **PB-1, 2**, and **3** could be determined in the hemolysis assay. Biolayer interferometry (BLI), however, still indicated the binding of **PB-2** to PLY-E151Q, PLY-E151K, and PLY-D205R (Supplementary Fig. 12g). From these data, it can be concluded that the

binding site of **PB**-molecules is not in the oligomerization domain. Consequently, two PLY mutants with mutations in the second potential binding site, the cholesterol-binding domain, were constructed. The PLY double mutant (PLY-DM, T459A, and L460A) with the two amino acids of the cholesterol recognition motif (CRM) modified was inactive in hemolysis, as expected for deleted cholesterol binding[11]. In the BLI assay, however, **PB-2** was still able to bind the PLY-DM mutant (Supplementary Fig. 12c, g). Thus, reconsidering that the olefinic double bond in all **PB**-compounds rendered them potential Michael acceptors, we decided to mutate the cysteine residue within the undecapeptide of the cholesterol-binding domain and expressed the PLY-C428A protein. This cysteine deletion mutant was active in the hemolysis and in the LDH assay and displayed similar cellular toxicity as wild-type PLY. **PB-1, 2**, and **3**, however, were unable to block PLY-C428A at concentrations completely inhibiting the wild-type protein (Supplementary Fig. 14c and Fig. 3a). In the BLI assay, **PB-3** generated on sensor surfaces with immobilized PLY-C428A at all tested concentration (2, 4 and 8 μM) no response in comparison with unloaded, negative control sensor, indicating that there was no association with the mutant protein (Fig. 4c). All these findings suggested that **PB-3** binds to Cys428 in the cholesterol-binding domain of wild-type PLY for inhibition, according to the kinetic data from BLI as a reversible, due to the low off-rate most likely covalent ligand.

To confirm this binding mode of **PB-3** protein mass spectrometry was conducted. Wild-type PLY provided a deconvoluted the calculated protein mass at 55536 Da as the main peak, whereas the mutant protein PLY-C428A exhibited a deconvoluted mass peak at 55504 Da, displaying the expected mass difference of one sulfur atom, 32 Da. A 1:1 mixture of **PB-3** with mutant PLY-C428A (6 μM each) delivered no change in the protein MS, while wild-type PLY delivered a deconvoluted protein spectrum with a broad protein mass peak (from 55558 Da to 56015 Da) ranging from the protein mass to the protein mass plus inhibitor **PB-3** (Supplementary Fig. 15). Finally, a putative structure of the complex of **PB-3** with PLY protein was generated by covalent docking. The approach indicated the accessibility of the electrophilic double bond for Michael addition of Cys428 and suggested binding of **PB-3** through hydrophobic interactions and H-bonding with the amino acid residues of domain 4 (Fig. 4f).

## Protein-templated formation and specificity of PB-3

During recent years several very selective Michael acceptors have been successfully introduced in the clinic including blockbuster drugs such as ibrutinib, afatinib, or neratinib, all binding covalently to non-activated Cys-residues[39]. Therefore, the selectivity of **PB-3** for wild-type PLY was evaluated by several approaches. First, a diverse collection of 12 Michael acceptors which had been reported previously to target cysteine residues in other proteins[40] or were derivatives of **PB-3** was investigated as potential inhibitors of PLY in the hemolysis assay. None of these compounds inhibited PLY (Supplementary Fig. 14a, b). Secondly, **PB-3** was tested with two enzymes containing activated nucleophilic cysteine residues, the cysteine protease SARS-CoV2 main protease and cysteine-containing protein tyrosine phosphatase PTP1B without detecting inhibition (Supplementary Fig. 13d). Deactivation of **PB-3** by inactivated cysteine residues was not observed either, as indicated by the activity of **PB-3** in the cellular assays, which were performed in Ham´s F12 medium containing 200 μM of L-cysteine, a 13,000-fold excess of free cysteine relative to the target protein PLY.

For comparison, the effects of PLY inhibitor **PB-3** on the activity of the human pore-forming toxin perforin were investigated. Recently, a set of small molecule inhibitors of perforin was reported[41], the structures of these molecules were, however, unrelated to **PB-3**. For a functional assay of perforin activity, human natural killer (NK) cells were isolated and were employed to lyse K562 target cells, a process mediated by the release and activity of perforin and granzymes from cytolytic granules contained within NK cells[42]. **PB-3** did not inhibit the

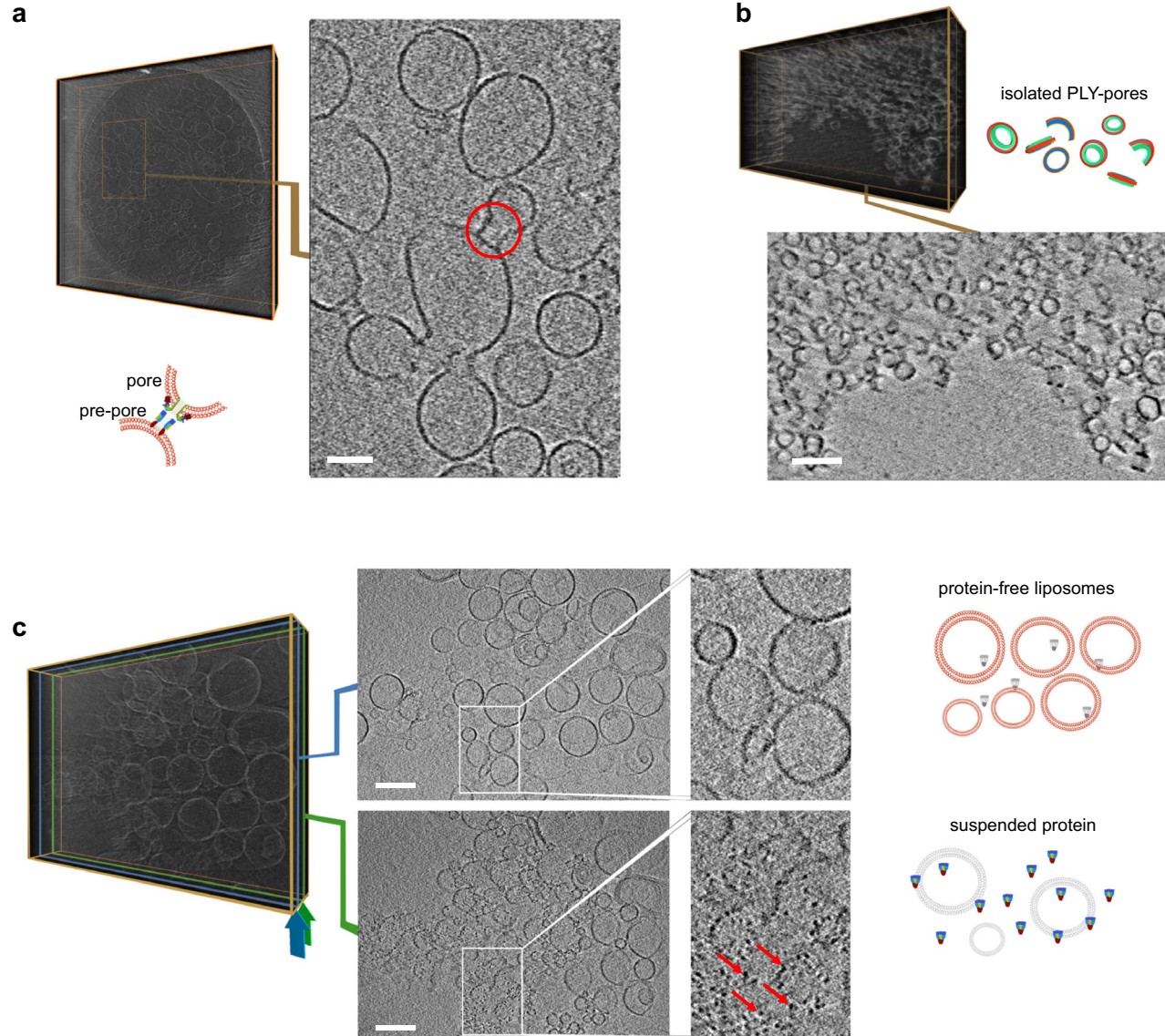

**Fig. 3 | Investigation of PLY and PB-2 using Cryo-TEM. a** Cryo-electron tomography of liposomes treated with PLY: The right image shows a section of a central stack of 20 accumulated slices from the reconstructed 3D volume (left, as *voltex* presentation), one example of PLY-induced pre-pore and pore formation event[38] is highlighted by a red circle. For a video representation see Supplementary Video 1. **b** Cryo-electron tomography of PLY-pores: Top: Reconstructed 3D volume of solubilized liposomes. Bottom: Central accumulated stack of 20 slices from the 3D volume showing protein pores formed by oligomerization of PLY monomers isolated by treatment of PLY-liposome mixture with Cymal-6 followed by Amphipol-35[12]. For a video representation see Supplementary Video 2. **c** Cryo-electron tomography of liposomes in the presence of PLY and inhibitor (**PB-2**): A central stack of 20 accumulated slices (position corresponds to the blue frame in the 3D volume on the left) shows smooth vesicle membranes (top image and enlarged detail). A near-surface stack of 20 accumulated slices (position corresponds to the green frame in the same 3D on the left) shows suspended PLY particles not integrated in liposomal membranes (highlighted by red arrows in the bottom image and enlarged detail). For a video representation see Supplementary Video 3 (data shown in (**a**–**c**) are one of *n* = 3 and scale bars correspond to 100 nm).

lysis of K562 cells by NK cells derived from three individual donors at 1 and 3 μM concentration. At a concentration of 6 μM, which blocked PLY-activity completely, median target cell lysis was reduced moderately by 15.6% compared to the DMSO control, however, this effect was associated with reduced release of cytolytic granules by NK cells indicating that **PB-3** is not an inhibitor of perforin (Supplementary Fig. 16).

As another experiment substantiating the specific interactions between the protein and its inhibitor, we investigated whether PLY protein catalyzed the formation of **PB-3** in a protein-templated reaction[40,43–45]. The two precursor fragments of **PB-3**, aldehyde **PB-3.1** and barbiturate **PB-2.3**, did not inhibit PLY alone. When, however, fragments **PB-3.1** (12.5 μM) and **PB-2.3** (50 μM) were incubated with PLY (15 nM) in PBS at room temperature for 1 h, about 50% inhibition

of PLY was observed in the hemolysis assay (Fig. 2b). In contrast, a solution of both fragments incubated in buffer without PLY protein did not result in the inhibition of hemolysis by PLY. These results suggested the protein-templated formation of the PLY inhibitor **PB-3** in the presence of the target protein but not in buffer without protein. To quantify the protein-templated reaction HPLC-QTOF-MS was used. Solutions of **PB-3.1** (50 μM) and **PB-2.3** (100 μM) were incubated with PLY (5 μM) and without protein in PBS and analyzed over 7 h (420 min). Formation of **PB-3** was detected only in protein-containing samples and was saturated (Fig. 2c). Using a calibration curve, molar concentrations of **PB-3** were determined and were fitted to a one-phase saturation curve $c(t) = c_0 + (c_{max}-c_0)*(1-e^{-K*t})$. Templated formation of **PB-3** by PLY protein yielded a maximal concentration of 795 nM with a reaction half-time ($t_{1/2} = \ln2/K$) of

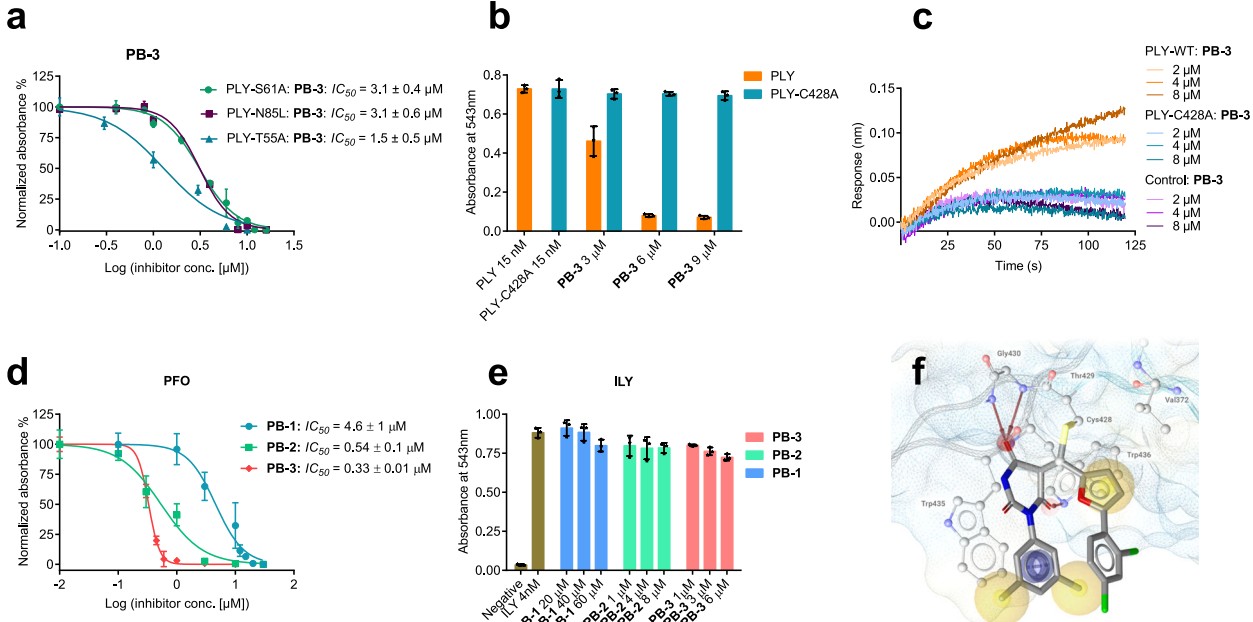

**Fig. 4 | Investigation of the biological activity and binding interactions of PB-inhibitors against PLY-mutants and evaluation of PB-inhibitors against alternate CDC. a** $IC_{50}$ values of **PB-3** against PLY-S61A, PLY-N85L, and PLY-T55A, which are mutants involving the potential binding site at the oligomerization interface (distinct samples $n = 3$, 95% CI for $IC_{50}$ and error bars represent ± S.D.). **b** Comparison of **PB-3** activity against wild-type PLY and PLY-C428A (a UDP-cysteine mutant version of PLY). **PB-3** blocks only wild-type PLY and is unable to neutralize the mutant PLY-C428A at all three concentrations (distinct samples $n = 3$, error bars correspond to ± S.D.). **c** Bio-layer interferometry assay: wild-type PLY, PLY-C428A-immobilized, and control NiNTA biosensors exhibit association with **PB-3** at three concentrations (2, 4, 8 μM). Only PLY (wild-type) shows significant association with **PB-3** in contrast to PLY-C428A (data presented are one of $n = 3$ distinct samples). **d** $IC_{50}$ values of **PB-1**, **2**, and **3** against perfringolysin (PFO, a toxin homologous to PLY) and **PB**-inhibitors show 10-fold increased potency toward PFO than PLY (distinct samples $n = 3$, 95% CI for $IC_{50}$ and error bars represent ± S.D.). **e** Intermedilysin (ILY, a cysteine-free CDC toxin) is not blocked by **PB**-inhibitors (distinct samples $n = 3$ for each experiment, 95% CI for $IC_{50}$ and error bars represent ± S.D.). **f** Covalent binding mode of **PB-3** suggested docking, for details see text.

21 min. The observed maximal concentration of **PB-3** is in good agreement with the binding affinity ($K_D$) of the inhibitor and reflects the saturation of the binding site being no longer available for a continued templated reaction[46].

### Activity of PB-3 toward other CDC

Having identified the binding site of **PB-3** at Cys428 within the UDP sequence of the cholesterol-binding domain and realizing that the UDP sequence is essential for the activity of many CDC we wondered whether **PB-3** would be active against other CDC as well. To address this question, two alternative CDC were expressed and investigated with **PB** molecules. Perfringolysin O (PFO), the CDC virulence factor of *Clostridium perfringens*, was selected as an example for CDC containing cysteine in the UDP sequence. Complete hemolysis of erythrocytes was effected at 1 nM concentration of PFO and all **PB**-compounds were potent inhibitors of PFO with about 10-fold decreased $IC_{50}$ values for **PB-1** (4.6 μM), **-2** (0.5 μM) and **-3** (0.3 μM) toward PFO compared to PLY (Fig. 4d). Intermedilysin (ILY), the CDC of *Streptococcus intermedius*, was finally selected as a negative control, having no cystein in the UDP sequence and displaying an alternative mechanism of pore formation which relies on the interaction with human CD59 protein, not with cholesterol. As a result, ILY effects hemolysis not of sheep but only of human erythrocytes at 4 nM concentration. None of the three **PB**-compounds inhibited hemolysis exerted by ILY again confirming the necessity of the UDP-cysteine for binding and activity of **PB-3** (Fig. 4e).

### PB-3 protects human lung cells from wild-type PLY during infection with *Streptococcus pneumoniae*

The protective effect of **PB-3** against PLY in A549 cells was illustrated by using confocal laser-scanning microscopy (Fig. 5, results with **PB-2** are shown in Supplementary Fig. 9). Mitochondria of cells were labeled with tetramethyl-rhodamine ethyl ester (TMRE) and a fluorogenic tetrapeptide substrate to visualize the activity of caspase-3/7 and to monitor apoptosis induction. Cells treated with PLY only, with PLY together with **PB-3** (6 μM), with **PB-3** only, and untreated healthy cells were incubated for 24 h, and fluorescent micrographs were recorded (Fig. 5a). **PB-3** was not only harmless to cells, rather it effectively protected cells from PLY-associated cellular damage.

Next, the effects of **PB-3** were investigated in an in vitro cellular-bacterial infection model. A549 cells were infected with a capsid-deletion mutant of *Streptococcus pneumoniae (S.pn.)* **D39Δcps**[47]. In this bacterial strain the capsid gene was deleted to ensure that cellular damage predominantly depended on PLY. In cells infected with **D39Δcps**, after 16 h of incubation approximately 100% of the cells were killed as indicated by the loss of mitochondrial TMRE signal, strong caspase activation as well as morphological signs of cell damage (Fig. 5b, II). In contrast, cells infected with **D39Δcps** treated four times after regular intervals of 4 h with **PB-3** (6 μM) almost entirely survived and did not show signs of apoptosis (Fig. 5b, III). Cells infected with a second mutant of *S.pn.* (**D39ΔcpsΔply**) carrying an additional deletion of the PLY-gene were regarded as negative controls without PLY-related damage. **D39ΔcpsΔply**-infected cells did not show signs of cellular damage or apoptosis suggesting that the observed cellular damage observed before indeed resulted from the action of PLY.

## Discussion

In this study, we have demonstrated the blocking of cholesterol-dependent cytolysin (CDC) and virulence factor pneumolysin (PLY) of *Streptococcus pneumoniae* by targeted small molecule inhibitors **PB-1**, **PB-2**, and **PB-3**. Inhibition of PLY blocked hemolysis of erythrocytes and protected human alveolar cells against PLY. **PB-2** inhibited membrane insertion of PLY and the formation of PLY pores as visualized by

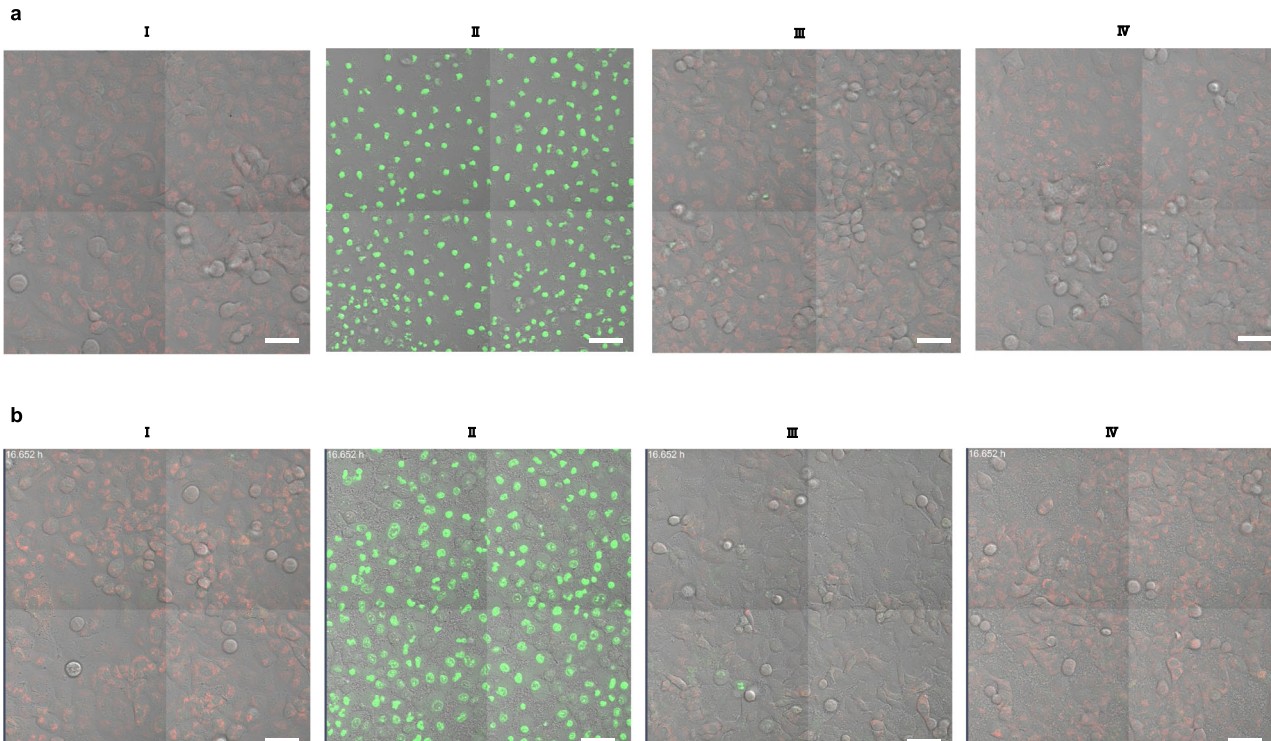

**Fig. 5 | Inhibition of PLY in native and *S.pn.*-infected human alveolar epithelial cells.** PLY neutralized by **PB-3** in human alveolar epithelial cells (A549 cells). Cells were grown in Ham's F12 culture medium with inclusion of a red mitochondrial dye (TMRE: tetramethyl-rhodamine, ethyl ester) to indicate healthy cells and a fluorogenic peptide substrate (DEVD) with green fluorescence to highlight activity of caspase 3 and 7 activity in apoptotic cells (data here represent one of $n = 3$, distinct samples). **a** (I) Healthy cells as a negative control, 24 h; (II) cells treated with PLY 15 nM, 24 h; (III) cells treated with PLY (15 nM) + **PB-3** (6 μM), 24 h; (IV) cells with **PB-3** as a control (24 h). **b PB-3** blocked cellular deterioration in A549 cells infected with *S.pn.* D39 strain (data shown is one of $n = 3$, distinct samples). (I) healthy control cells, (II) cells infected with **D39Δcps** induced cellular injury (16 h), (III) cells infected with **D39Δcps** treated with **PB-3** (6 μM) 4-times after regular intervals of 4 h (16 h), and (IV) **D39ΔcpsΔply** control, and it represents that only PLY provokes the cellular injury. All images were recorded by laser confocal microscopy after the mentioned time intervals. Scale bars correspond to 50 μm.

cryo-transmission electron tomography of cholesterol-containing liposomes and resulted in protein precipitation. Chemically optimized inhibitor **PB-3** bound wild-type PLY reversibly with an affinity ($K_D$) of 256 nM, resulting from a rapid on-rate and a slow off-rate, as determined by bio-layer interferometry. The observed binding kinetics corresponded to a residence time of the protein-ligand complex of >2000 s. Mutational analysis of PLY protein using eight mutants in two potential binding domains identified Cys428 in the undecapeptide sequence of the cholesterol binding domain as the binding site of **PB-3**. This finding excluded the binding site in the oligomerization domain addressed in virtual screening and emphasized the need for a careful experimental validation of the binding sites of small molecules derived from virtual screening. The long residence time, the deletion of binding as well as inhibition of the mutant PLY-C428A, and the electron-deficient olefinic double bond being essential for activity suggested a covalent reversible mode of action of **PB-3** as PLY inhibitor. This mode of action was supported by a plausible structure of the covalent inhibitor-protein complex derived from covalent docking. Most importantly, **PB-3** protected human lung epithelial cells against PLY-mediated cell death during infection with *Streptococcus pneumoniae*. While untreated cells in the cellular disease model were fully apoptotic after 16 h, **PB-3** effectively blocked apoptosis induction of PLY-treated as well as infected cells. During treatment of the bacterial infection, repetitive dosing of **PB-3** was required, which indicated the continuous production of PLY by the bacteria and, possibly, the limited metabolic stability of **PB-3**.

In summary, this work established CDC toxins like PLY as small molecule protein targets by using Cys428 within the UDP sequence of the cholesterol-binding domain as binding site. Michael acceptors like **PB-3** fulfill essential selectivity and stability criteria of covalent drugs[39,46] and thus can be considered as lead structures for the pre-clinical development of PLY inhibitors, which might constitute an effective anti-virulence strategy against PLY-mediated pathogenesis. This strategy should be extendable to other homologous CDC also containing Cys in the undecapeptide sequence, like perfringolysin (PFO). Such advanced CDC inhibitors in the future might serve as adjuncts of antibiotics or even replace them for the prevention of invasive CDC-related infections possibly opening a new window of therapeutic opportunities.

## Methods
### Computational methods
A structural model of a PLY dimer was generated by mapping atomic coordinates from the monomer crystal structure 4ZGH[30] into the cryo-EM map 2BK2[31] containing Cα coordinates using the alignment tool in MOE (Molecular Operating Environment (MOE), 2019.0102; Chemical Computing Group Inc.). We defined a size and shallowness to host drug-like molecules and the proximity to the dimer interface as prerequisites for a potential binding site and applied Site Finder (MOE) to identify such a pocket. Virtual screening of AnalytiCon and Specs libraries containing 554.711 molecules was carried out with the CCDCs software GOLD version 5.1 in the virtual screening mode by using Goldscore as primary scoring function[48]. Resulting hit molecules with a Gold fitness score above 40 were filtered by molecular weight (200–500 μ) and clogP values (−0.4 to 4). All resulting hit molecules were re-scored with a pharmacophore model generated by frequent

interactions found in the 50 highest-scored molecules. After clustering by pharmacophore similarity (3 conformations, cluster distance 0.4, average) and visual inspection, 10 compounds were selected for biological testing in the hemolysis assay. All pharmacophore applications were performed with LigandScout 4.4.3[33]

## Method of covalent docking
PLY crystal structure 4ZGH[30] was prepared using Molecular Operating Environment 2022.2 (Chemical Computing Group, Montreal, Canada) using Protonate3D[49]. The covalent docking protocol implemented in MOE was employed to generate docking conformations attached to to C428. Resulting docking poses were subsequently minimized using the MMFF94 implementation of LigandScout 4.4.3[33] and ranked according to their interaction patterns as analyzed by LigandScout.

## Mutagenesis, expression, and purification of recombinant pneumolysin variants
Established protocols of PLY expression were customized in this study[10,12,30]. A cDNA encoding the wild-type sequence of PLY was cloned between the NdeI and the XhoI site of the plasmid pET-19b (Novagen). Mutants of wild-type PLY had been prepared with the wild-type cDNA as template by site-directed mutagenesis. PCR was performed using Pwo SuperYield DNA Polymerase, dNTPack (Roche Diagnostics GmbH, Germany) according to the manufacturer's procedure. Plasmids containing the mutant PLY gene were purified by QIAprep®Spin Miniprep Kit (Qiagen, Germany). Primer sequences are available in Supplementary Table 3. Transformed E.coli (BL21) were grown in Luria Broth (LB) medium at 37 °C till the optical density reached 0.8, further expression was induced after addition of 0.1 mM IPTG at 25 °C, after 4 h bacteria were centrifuged and stored at −20 °C. Bacteria were resuspended in lysis in Buffer (25 mM Tris, 500 mM NaCl, 20 mM imidazole, 10% glycerol, 2 mM β-ME, and pH 7.2) containing 100 μg/ml lysozyme, and 100 μM PMSF. Bacteria were further broken down by sonification cycles (10×) each cycle lasted 20 s. Cellular debris was removed by spinning down. Further, PLY was extracted by using 2 ml column of HisPur Ni-NTA Superflow Agarose. The Ni-NTA column was equilibrated by buffer (25 mM Tris, 500 mM NaCl, 10 mM imidazole, 10% glycerol, 2 mM β-ME, and pH 7.2) and the protein mixture was flushed through the column. Then, the column was washed with buffer (25 mM Tris, 500 mM NaCl, 25 mM imidazole, 10% glycerol, 2 mM β-ME, and pH 7.2) and PLY was eluted by buffer (25 mM Tris, 500 mM NaCl, 300 mM imidazole, 10% glycerol, 2 mM β-ME and pH 7.2). PLY was additionally purified on HiLoad Superdex 75 pg preparative size exclusion chromatography column with buffer (20 mM MOPS buffer, 5% glycerol and pH 8) (ÄKTA pure with UNICORN V6.3 software, GE Healthcare) The PLY fractions were combined, concentrated, analyzed on LCMS (Supplementary Fig. 17, 18, 19, 20, and 21) and SDS-gel, then stored at −80 °C. The protein concentration was measured with NanoDrop One (Thermo Fischer Scientific) using NanoDrop One Software V1.4.2. The expression and purification of wild-type PLY as well as its mutant variants were carried in similar manner.

## Hemolysis assay
Hemolytic activity of toxins was described in multiple studies[11,25]. Defibrinated sheep blood was purchased from Thermo Scientific. Blood was centrifuged at $3500 \times g$ and serum was removed. Pelleted erythrocytes were resuspended in the same volume of PBS buffer (137 mM NaCl, 2.7 mM KCl, 10 mM $Na_2HPO_4$, 0.24 mM $KH_2PO_4$, and pH 7.4) and centrifuged again. This washing procedure was conducted three times. Finally obtained erythrocytes were resuspended in same volume of PBS and aliquoted for later use. Inhibitors were dissolved in DMSO stock solutions (5–20 mM) and were diluted in PBS buffer (pH 7.4) at different concentrations with the final DMSO concentration always below 1.5% (v/v). Subsequently, PLY solution

was added to a final concentration of 15 nM or 0.8 μg/ml and the mixture was incubated for 10 min at 37 °C. PBS alone, PLY alone, and 10% Triton X-100 samples were used as controls. In the next step, sheep erythrocytes (2.5% of total assay volume, ca. $1 \times 10^7$ erythrocytes per μl) added to each sample and incubated for 15–20 min at 37 °C. Human serum was purchased from Pan-Biotech Germany and serum containing volume. Followed by incubation, all samples were centrifuged at $9000 \times g$ for 2 min. Supernatants of every sample were isolated in a 96-well plate and absorption was measured at 543 nm in a spectrometer (Tecan) using Magellan V7.2 data acquisition software. The data were normalized and $IC_{50}$ as well as $LD_{50}$ values were calculated by "log (inhibitor) vs. normalized response-variable slope", and "log (conc. of PLY) vs. normalized response-variable slope" curve fittings using GraphPad Prism 6 (software), respectively.

## Quantitative analysis of PB-3 formation via protein-templated fragment ligation
The experiments were performed using an Infinity II 1290 HPLC (Agilent Technologies) coupled with an Agilent 6550 iFunnel Q-TOF/MS. A calibration curve of **PB-3** was established over 0.312–10 μM. To quantify the fragment ligation product, the fragments **PB-3.1** (50 μM) and **PB-2.3** (100 μM) were incubated in PBS with PLY (5 μM) and without protein (control) at room temperature. The final DMSO concentration in samples was ≤7% and, likewise, the final glycerol concentration was ≤3%. The samples were first analyzed after 0.5 h and, then, after each h up to 7 h. The fragments were dissolved into PBS from DMSO stock solutions and protein was dissolved from buffer: 20 mM MOPS, 50% glycerol with pH 8. HPLC column: Zorbax eclipse plus C18 (1.8 μm 2.1 × 50 mm). HPLC parameters: injection volume: 1 μl, gradient: 0–8 min from 95/5 (A/B) to 5/95 (A/B), 8–9 min 5/95 (A/B), 0–1 min waste. QTOF parameters: fragmentor 175 V, nozzle voltage 1000 V, Vcap 4000 V. The data was later analyzed by EIC/Find by Formula algorithm on Agilent Masshunter Qualitative Analysis v. 10.0 (software). The unknown values were determined via calibration curve of **PB-3** by using GraphPad Prism 6. Next, the quantified values of the product (**PB-3**) were plotted vs time, and nonlinear regression one-phase association equation was applied [$Y = Y0 + (Plateau-Y0)^*(1-exp(-K^*x))$] using GraphPad Prism 6.

## Kinetic analysis of inhibitors via bio-layer Interferometry (BLI)
To calculate binding kinetics between PLY and PB-molecules (**PB-2** and **PB-3**), FortèBio Octet K2 system and FortèBio Ni-NTA biosensors were employed. The experiment was executed in a black flat bottom 96-well plate and biosensors were equilibrated in recommended kinetic buffer (PBS, 0.02% tween 20, 0.05% $NaN_3$) for 1 h before proceeding. All DMSO stock solutions of inhibitor and reference (DMSO) were dissolved in kinetic buffer, the total concentration of DMSO was with less than 5%. PLY and PLYC428A, the latter used as a negative control and both tagged with a 10x-histidine were diluted in PBS with a final concentration 100 μg/ml. At room temperature, all samples including PLY, PLYC428A (reference), multiple concentrations of inhibitor, and respective buffer control samples were transferred to 96-well plate, a working volume of 200 μl was used for each well. The layout of assay was designed in FortèBio data acquisition9 software, the experiment was carried out in following steps: 1st initial baseline for 60 s, 2nd protein loading for 400 s, 3rd inhibitor baseline for 60 s, 4th association of inhibitor for 120 s and 5th dissociation for 200 s. Last three steps (3rd, 4th, and 5th) were repeated for each concentration of inhibitor. After completion, experimental data were analyzed utilizing FortèBio data analysis9 software, reference data (protein, inhibitor, and buffer) was subtracted from the raw data and according to baseline aligned. Global fitting of 1:1 kinetic model was applied to determine binding kinetics and statistical errors.

## Investigation of PB-3 binding to PLY and PLYC428A by BLI

The association comparison between PLY and PLYC428A with **PB-3** was investigated on FortèBio Octet K2 system and FortèBio Ni-NTA biosensors were used. Both proteins with 100 µg/ml concentration were immobilized to adjacent biosensors. After inhibitor baseline, protein biosensors were stepwise soaked in wells containing different concentrations of **PB-3**. At the end of experiment, the association-step raw data for both proteins against all concentrations of **PB-3** was baseline aligned in FortèBio data analysis9 software, next the data transferred to GraphPad Prism 6 and plotted on a graph.

## Evaluation of PB-molecules binding to PLY-mutants and PLY-WT

BLI assay was selected and **PB-2** was chosen for the qualitative analysis of interaction between PLY-mutant versions (PLYD205R, PLYE151Q, and PLY-DM) and PLY wild-type. Equal concentrations (~100 µg/ml) of Mutant version and PLY-WT were loaded on parallel Ni-NTA biosensors. Next, the inhibitor baseline was achieved and, for determining association and dissociation, protein-coated sensors were systematically immersed in wells containing different concentrations of **PB-2**. The same experiment repeated for three mutant versions of PLY. Later, raw data of association-dissociation steps of each experiment, at 4 µM of **PB-2**, was highlighted and transferred from FortèBio data analysis9 software to GraphPad Prism 6. Finally, data of mutant versions and PLY-WT was combined in a graph, and protein interaction with **PB-2** (4 µM) was highlighted as association and dissociation steps.

## Lactate dehydrogenase (LDH) assays

The LDH assay was chosen because the release of LDH provides an indication of membrane rupture upon pore formation by PLY, whereas other assays such as MTT or Alamar Blue could only quantify the reducing potential of cells. To perform these assays, we purchased Pierce LDH Cytotoxicity Assay Kit from Thermo Scientific for 1000 reactions. We had grown the human alveolar epithelial cells (A549 cells) (ATCC, Cat#. CCL-185) in Ham's F12 medium complemented with 10% FCS (fetal calf serum). We used 12-well plates and cultivated with $1 \times 10^5$ cells per well. Next, PLY (15 nM)+ increasing concentrations of inhibitors were incubated for 4 h and 24 h (h = hours) at 37 °C. Cells with media and 5% triton X-100 samples were kept as negative and positive controls, whereas PLY alone and inhibitor (highest concentration) alone samples were regarded as PLY and inhibitor controls respectively. After incubation time, cellular supernatants of each sample were removed, centrifuged at $400 \times g$ for 10 min at 4 °C.

LDH reagent from the reaction kit was mixed with supernatant in 1:1 ratio in a 96-well plate. This mixture was left in dark for 30 min, later absorption at 490 nm was measured in a microplate reader. The cytotoxicity of each sample was quantified in percent by using the formula: Cytotoxicity (%) = (Experimental value − Negative control) × 100 / (Positive control − Negative control). Graphs were plotted with Graphpad Prism 6.

## Liposomes preparation

Liposomes were synthesized by cold extrusion. A mixture of 70% DOPC (1,2-dioleoyl-sn-glycero-3-phosphocholine) 30% cholesterol was prepared in chloroform, next, the solvent was evaporated by rotatory evaporation at 40 °C with 475 mbar, followed by 0 mbar, led to formation of thin film of dried lipids. In the next step, the dried lipids were rehydrated in PBS buffer and the suspension was overnight shaken at 500 RPM at room temperature[11,12,31]. After that, 4 freeze-thaw cycles were carried out in liquid nitrogen. A polycarbonate filter (100 nm) was applied to extrude liposomes over a manual extruder (Avanti polar lipids. USA) in 25 cycles at room temperature[12]. The liposomes were characterized by dynamic light scattering (DLS) (Nicomp Z3000) with a mean diameter of 101.7 ± 27.7 nm and polydispersity index of 0.074,

at 23 °C with an intensity of 300 KHz. DLS data was acquired and analyzed by Nicomp Z3000 Software.

## Protein-liposomes sample preparation for Cryo-TEM

To prepare protein-liposome sample, liposomes (2 mM, 70:30 DOPC: Cholesterol) were incubated with PLY (1 mg/ml) for 1 h, then Cymal-6 (0.56 mM) was added and 10 min further incubated at 37 °C, after this sample was ready for application on to the Cryo-TEM grid. In case of protein-inhibitor and liposomes, samples were prepared in exact same manner except, PLY (1 mg/ml) was preincubated with **PB-2** (100 µM) for 10 min. Liposomes (2 mM) alone sample in PBS was kept as control (n = 3). To isolate PLY-induced pores, in the mixture of liposome (2 mM), PLY (1 mg/ml), and Cymal-6 (0.56 mM) after their respective incubation time, amphipol was added in a ratio (w/w) 5:1 to PLY[12]. This mixture was incubated for 30 min at room temperature. Before applying the sample to Cryo-TEM grid, detergents were exchanged with PBS by dialysis cassette (Slide-A-Lyzer® G2, Thermo Scientific, USA). Buffer was exchanged for 2 h and again 2 h, then overnight at 4 °C.

## Cryo-TEM investigation

**Preparation of cryo-TEM-samples.** Samples were prepared by use of a Vitrobot™ Mark IV (Thermo Fisher Scientific). The Vitrobot was operated at 22 °C and at a relative humidity of 100%. In the preparation chamber of the Vitrobot, a 4 µl sample was applied on a Quantifoil (R4/1 batch of Quantifoil Micro Tools GmbH, Jena, Germany) grid which was surface plasma treated for 60 s at 10 mA just prior to use. Excess sample was removed by blotting for 4 s, and the ultrathin film thus formed was plunged into liquid ethane just above its freezing point.

**Imaging of cryo-TEM-samples.** Vitrified samples were imaged with a Talos Arctica™ TEM (Thermo Fisher Scientific) at 200 kV accelerating voltage and a primary magnification of 28 k. Images were recorded by a Falcon III direct electron detector at full 4 k resolution resulting in a pixel size of 0.373 nm/pixel. The defocus value was set to 4.9 µm.

## Cryo-electron tomography

Vitrified samples were transferred under liquid nitrogen into a Talos Arctica™ TEM (Thermo Fisher Scientific, Hillsboro, Oregon; USA) at 200 kV accelerating voltage employing a Volta phase plate in the back focal plane of the objective lens at a primary magnification of 28 k. Image series in the tilt range of −64°/64° (2° tilt-increments) were recorded by a Falcon III direct electron detector at full 4 k resolution resulting in a pixel size of 0.373 nm/pixel. The defocus value was set to −300 nm. A total dose of 180 e/Å$^2$ was accumulated on the specimen. The image stack alignment, 3D reconstruction, and data processing were performed in the context of the FEI Inspect 3D software V4.3 (Supplementary Table 2).

## Human lung cells protection from native PLY

The human lung alveolar epithelial cell line A549 (ATCC, CCL-185) was cultured in Ham's F12 medium (Biochrome, Berlin, Germany) supplemented with 10% (v/v) fetal bovine serum (FBS; Capricorn Scientific GmbH, Ebsdorfergrund, Germany) and 2 mM L-glutamine (Invitrogen Life Technologies) at 37 °C and 5% CO2. A549 cells were seeded in 8-well-ibidi-slides (25,000 cells per well), after 24 h the medium was changed to Ham´s F12, 2% FBS. Next day membrane potential was determined by loading cells with 10 nM tetramethyl-rhodamine methyl ester (TMRE from Life Technologies) for 30 min at 37 °C/5% CO$_2$ and washing 3 times with Hanks' balanced salt solution (HBSS). Then the Medium was changed to Ham´s F12 with CellEvent Caspase 3/7 Green Detection Reagent (Life Technologies), a fluorogenic peptide substrate of caspases 3 and 7. Stimulation was made with 15 nM PLY, with PLY preincubated with inhibitor for 10 min (**PB-2**: 8 µM, **PB-3**: 6 µM) and with inhibitor alone.

## Streptococcus pneumoniae infection mitigation

The human lung alveolar epithelial cell line A549 (ATCC, CCL-185) were seeded and grown, as mentioned earlier. In the second step, A549 cells after staining with TMRE were infected with the *S. pneumonia*. The pneumococci strains were provided by Prof. Dr. Sven Hammerschmidt from Universität Greifswald[50]. D39 derived capsule locus (*cps*) deletion mutant D39Δ*cps* and the *cps*/pneumolysin double mutant D39Δ*cps*Δ*ply* cultured in Todd-Hewitt broth supplemented with 0.5% yeast extract (BD) to midlog phase using a multiplicity of infection of 1 in Ham's F12 medium supplemented with 2% FBS for 24 h at 37 °C/5% $CO_2$. One well was infected with D39Δ*cps* and inhibitor **PB-3** (6 μM) was added. Addition of 6 μM **PB-3** was repeated after every 4 h for 16 h.

## Live-cell microscopy

It was performed on a LSM780 confocal laser-scanning microscope driven by Zen 2012 software using a 40×/1.2 NA C-Apochromat objective with 488 nm, 561 nm, and 622 nm excitation lines (Carl-Zeiss). Cells were imaged at 37 °C/5% $CO_2$ in phenol red-free medium for 24 h in case of PLY-inhibitor experiment and for 16 h in case of *S. pneumoniae* infection experiment.

## Protein-ligand complex investigation by denaturing MS

Mass spectra of the protein-inhibitor complex were compared to protein (control) by using an Infinity II 1290 HPLC (Agilent Technologies) coupled with an Agilent 6550 iFunnel Q-TOF/MS. Agilent 6550 iFunnel Q-TOF/MS. A mixture of 1:1 ratio of protein to inhibitor (6 μM, both) was prepared in buffer (50% acetonitrile + 50% water) with a final volume of 20 μl, samples contained up to 5% glycerol and up to 6% DMSO. Protein-inhibitor and protein-alone samples were measured after 2–5 min of incubation. The final data was analyzed with help of Agilent MassHunter BioConfirm 10.0 Software. The deconvoluted spectra were integrated via maximum entropy-algorithm with no limited m/z-range. HPLC parameters: Jupiter C18- column (5 μM, 300 Å, 3 × 250 mm) Phenomenex, injection volume: 1 μl, gradient: 0–4 min 95/5 (A/B), 4–10 min from 95/5 (A/B) to 40/60 (A/B), 10–12 min 40/60 (A/B), 12–15 min from 40/60 (A/B) to 95/5 (A/B); QTOF parameters: fragmentor 350 V, nozzle voltage 2000 V, VCap 3500 V, 0–4 min waste.

## Investigating the inhibition of perforin-mediated cell lysis by NK cells

**NK cell isolation.** Peripheral blood mononuclear cells (PBMCs) were obtained from the peripheral blood of healthy donors (*n* = 3) through density gradient centrifugation after informed written consent (protocol approved by the Ethics Committee of the Ärztekammer Hamburg, reference number PV4780). NK cells were subsequently enriched from PBMCs using a negative-selection strategy employing the EasySep Human NK Cell Enrichment Kit (Stemcell). NK cells were washed and then resuspended in complete medium (RPMI-1640 medium) (Life Technologies) supplemented with 10% (v/v) heat-inactivated FBS (Biochrom), 100 U/ml penicillin, and 100 μg/ml streptomycin (Sigma–Aldrich) and rested overnight.

**Cytotoxicity assay.** After overnight resting, 50.000 NK cells were used against the K562 (were provided by the Altfeld laboratory and originally obtained from DSMZ, Cat.# ACC 10) target cell line in a flow-cytometry-based cytotoxicity assay at an effector to target ratio of 2:1. Concentrations of 1 μM, 3 μM, and 6 μM of **PB-3** were tested. 5 mM stock solution of **PB-3** in DMSO was used to generate a 200 μM solution in PBS, which was employed for tested concentrations. As a positive control for inhibition of NK cell lysis, 10 mM of ethylene diamine tetraacetic acid (EDTA) was used[51]. As vehicle control, 3 μl of 4% DMSO was used. Target cells were labeled with a fluorescent cell tracer dye (CFSE, Invitrogen) to measure surviving K562 cells at the end of the assay by flow cytometry. K562 cells were incubated with CFSE (1:1000)

for 8 min (mixing after 4 min) in 1 ml PBS at room temperature. Reaction was quenched by using 7 ml RPMI/10% FBS. NK cell:K562 cocultures were incubated for 3 h in 5-ml polystyrene round-bottom tubes (Corning) at a final volume of 100 μl at 37 °C, 5% (v/v) $CO_2$ in RPMI/10% FBS. After incubation, 25 μl of precision counting beads (BioLegend) were added to ensure uptake of equal volumes during flow cytometric analysis. CFSE+ cells were immediately counted at a BD LSRFortessa (BD Biosciences), and analyzed by FlowJo V10.8.1. Lysis was calculated using K562 without NK cells (K562 only) as a reference by following formula: 100- (K562 survived (absolute count)/K562 only (absolute count) × 100). Three individual donors were measured, with two technical replicates for each donor.

**Cell staining.** Following the initial target cell count, leftover samples from each condition were combined for a surface staining of degranulation and activation. Cells were transferred to 96 u-bottom plate (Sarstedt), washed with PBS, and stained with fixable Near-IR Dead Cell Stain (Invitrogen) as well as αCD56-BUV395 (BD Bioscience, clone: NCAM16.2, dilution 1:100), αCD16-BV711 (BioLegend, clone: 3G8, dilution 1:100), αCD107a-BV421(BioLegend, clone: H4A3, dilution 1:100), αCD69-PE (BioLegend, clone: FN50, dilution 1:100) in PBS supplemented with 2% FBS for 30 min at 4 °C. Cells were washed twice prior to intracellular staining. Perforin staining was performed using BD Cytofix/Cytoperm Fixation/Permeabilization kit (BD Biosciences) and αPerforin-APC (BioLegend clone:dG9, dilution 1:50) according to the manufacturer's protocol. Cells were fixed with 1×CellFIX (BD Biosciences) and stored at 4 °C in PBS until flow cytometry.

**Chemical synthesis.** The synthesis procedures of **PB**-molecules are available in the Supplementary Information file under Supplementary Note 1.

## Reporting summary

Further information on research design is available in the Nature Portfolio Reporting Summary linked to this article.

## Data availability

The authors declare that the data supporting the findings of this study are available within the article and its Supplementary Information files. All raw data supporting the findings of this study have been uploaded as a source data file, which is also accessible in Figshare via the link [https://figshare.com/articles/dataset/_A_b_b_Targeted_b_b_Small_Molecule_Inhibitor_with_Elongated_Residence_Time_Blocking_the_Cytolytic_Effects_of_Pneumolysin_and_Homogolous_Toxins_b_/25040261]. The MS data has been reposited as a part of the source data file and the Supplementary Information file under Supplementary Note 2. Crystallographic data of PLY monomers and Cryo-EM map of PLY pre-pore used in this study are available at Protein Data Bank under the following entries: 4ZGH and 2BK2, respectively.

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

## Acknowledgements

The authors are grateful for the support of this research by the Deutsche Forschungsgemeinschaft (DFG) in the framework of SFB 765, project B9, and SFB 1349, project A3, and the SFB-TR 84, projects B6 and Z1a. U.B.A.A. received a graduate fellowship from the SFB 765. We acknowledge the support from Charité-Zeiss MultiDim and from the Core Facility BioSupraMol of Freie Universität Berlin supported by the DFG. A.Hö. and T.T. were supported by funding from the Federal Ministry of Education and Research (01KI2110). A.A. received a full scholarship from the Ministry of Higher Education of the Arab Republic of Egypt (ID#2019/2020).

## Author contributions

J.R., G.W., A.C.H., A.Hoe, U.B.A.A., M.B., and A.S. conceived and designed the experiments, and U.B.A.A., M.B., M.M., A.S., A.A., C.B., and T.T. performed the experiments. U.B.A.A., A.S., K.L., and J.R. analyzed the data. C.A., A.A., and T.R. contributed reagents. J.R., U.B.A.A., A.C.H., A.S., M.B., and G.W. wrote the manuscript. J.R. acquired funding for and administered the project.

## Funding

## Competing interests

The authors declare no competing interests.
