## [Peer Review File · Nature Communications]

Targeted Small Molecule Inhibitors Blocking the Cytolytic Effects of Pneumolysin and Homologous ToxinsREVIEWER COMMENTS

Reviewer #1 (Remarks to the Author):

General comments:

This paper describes the design and development of small molecule inhibitors of pneumolysin (PLY), a protein from *Streptococcus pneumoniae*. PLY is a pore-forming toxin that is cytotoxic to host cells and further contributes to pathogenesis by enhancing bacterial colonization throughout the host. The aim of this study was to identify a small molecule that possessed the ability to neutralize the activity of PLY by blocking pore formation and might therefore be a potential candidate for further development as a human therapeutic.

The paper begins with a model derived from the crystallographic data of PLY monomers fitted into the cryo-EM map of the pre-pore. This enabled the identification of two putative binding pockets, one in the oligomerisation domain and one in the cholesterol binding domain. The former was chosen for this study due to its proximity to a salt bridge known to be crucial for (pre)-pore formation together with its location in an area of large-scale conformational change upon (pre)-pore formation. *In silico* docking of 555,000 compounds and refinement based on physicochemical properties and a pharmacophore model were carried out, giving ten scaffolds suitable for testing in an *in vitro* test of haemoglobin release from sheep erythrocytes. This resulted in the identification of the PB-1 series of PLY inhibitors. Two new generations of inhibitors, PB-2 and PB-3 were designed and synthesised and shown to prevent PLY-mediated damage of A549 (human lung) cells. Target engagement with PLY was shown by both bio-layer interferometry using PLY on a sensor surface, and inhibition of PLY pore formation in liposomes visualised by cryo-TEM and cryo-ET.

The binding site was then investigated further through mutational analysis of both putative pockets where it was found that the inhibitors bound not within the oligomerisation domain as expected, but rather the cholesterol binding domain. It was postulated that this was a covalent interaction due to the presence of a Michael acceptor (MA) in the hit compounds. This was confirmed in multiple ways; inactivity of an analogue without a MA, the inhibitors were unable to block PLY activity when the target cysteine was deleted, BLI showed a long residence time for PB-3 on the WT PLY and no association with the deletion mutant, and protein mass spectrometry confirmed the mass of PLY plus the ligand.

Protein templated formation of PB-3 from its constituent parts was demonstrated and specificity of the ligand investigated with a panel of MAs in the haemolysis assay where PB-3 appeared to have selectivity for PLY over other compounds in the panel. The selectivity of PB-3 for PLY over SARS-CoV2 and PTP1B (a human phosphatase), chosen as they contained activated nucleophilic cysteines, was also shown. The activity of PB-3 towards other CDCs was investigated and it was found to be a potent inhibitor of perfringolysin which contains a cysteine in the UDP sequence, but not intermedilysin, which lacks a cysteine in this position, further supporting a covalent interaction between the MA and this nucleophilic residue. Finally, it was shown using confocal laser scanning microscopy that PB-3 protected human lung cells from WT PLY during *Streptococcus pneumoniae* infection, and also from a capsid-deleted mutant where host cell damage is predominantly mediated by PLY – 100% apoptosis was observed in the absence of inhibitor, treatment with PB-3 resulted in virtually all cells surviving.

I would describe this paper as a mixed bag. The initial aim was to target inhibitors to the oligomerization domain as there was solid rationale for interfering with the pore-forming process. The *in silico* screening and subsequent refinement was based on this hypothesis. The hit list was filtered based on physicochemical properties (a standard approach), then described as being re-scored based on 'frequent interactions' – it is not clear from the manuscript what this actually means, nor is a reference given. Ten compounds were selected for *in vitro* testing, resulting in the identification of PB-1. A small number of structural analogues were then designed and prepared, however it is not described why these particular compounds were targeted. Molecular dynamic

simulations were done to identify a binding mode and interactions, (for some reason figure 1d only appears in the supplementary, not the manuscript) but this only showed interactions that could easily be identified by an experienced medical chemist. There is no real rationale given for the changes made, and very little discernable structure-activity relationship apart from the requirement for a MA. Additionally, some compounds in the experimental are not shown in Table 1. I note here that a series of small molecules containing a very similar MA that inhibit the pore-forming protein perforin have been published previously but are not referenced here (Lena et al 2008, DOI 10.1021/jm801063n). The instability of PB-1 and 2 is reported, but how the more stable PB-3 is arrived at is not explicitly described, clearly more compounds were made than just those reported here. And in the end, the compounds don't actually bind at the site that was originally targeted.

The PB compounds were then shown to prevent PLY induced damage in human lung cells and target engagement with PLY shown as described above. The actual binding site was then identified and characterised through an elegant series of mutagenesis experiments. Although the PB compounds appear to show selectivity (when compared against SARS-CoV2, PTP1B and a non-cysteine containing CDC), the compounds were not tested against human pore-forming proteins such as perforin. This would be a key experiment due to the possibility of off-target effects involving the immune system.

In summary, there is a disconnect between the initial aim and the actual outcome in this paper. Although a well-argued rationale was given for the target pocket and compound design, the study was essentially a very small in vitro screen of compounds that resulted in a modest inhibitor of PLY activity. Cellular experiments were carried out at a high concentration of inhibitor; 6 uM, single or multiple doses – the IC50 is 3.1 uM - it is not clear how test concentrations were determined? Inhibitors should also have been tested for off-target effects against CTL and/or NK cells to check if inhibition of the immune protein perforin, which possesses a similar pore-forming mechanism of action to the CDCs, affected lytic activity. The lipophilicity of the hit compound (PB-3 cLogP = 6.56) could be a significant driver of binding, leading to a host of off-target effects. Although the physicochemical properties are not reported, these compounds are unlikely to be very soluble and don't look much like 'drug-like' leads. While there are many very interesting and elegant experiments carried out to characterise these inhibitors, if the raison d'être of this paper is to identify 'lead structures for the preclinical development of PLY inhibitors' for eventual clinical use, then we are a long way from that point. Another approach to take with this story would be to relocate the medicinal chemistry/compound design into a separate paper (and add in any compounds that don't appear here) and in this paper focus on the current work where PB-2/3 were used as chemical tools to elucidate the mechanism of action for this scaffold.

More specific observations:

P3: Reference 26 appears before 19-25

P4: The statement 'however, these molecules have been reported active toward numerous proteins and against many diseases without adequate biological or clinical evidence' – is this the opinion of the authors or the reference given (26)? This statement needs further substantiation.

P5: How many compounds were tested in total? They don't all appear to be here.

P5: Fig 1d is referred to here, but is in the supplementary, not the main text.

Table 1: Compounds 2.6, 2.8, 2.10 are shown in the experimental but do not appear in the table with data. Compounds 2.7, 2.9, 2.11 (if they exist) do not appear at all.

P8: What compound concentrations were used in the 'PLY-inhibition protects human lung cells' A549 assay?

P8: 'Cellular lysis induced by PLY was quantified by determining the activity of lactate dehydrogenase (LDH) in each sample' Why this assay? The assay choice needs further explanation.

P8: Ref 38 appears before 37.

P11: It is not clear why the pore-blocking assay using cryo-TEM was carried out using PB-2; wouldn't it be better to use the more stable PB-3 as is done elsewhere?

P11: Refers to figure 3d, there is no 3d (means 3a?)

P13: The findings support or suggest, rather than establish binding to Cys428.

P14: 'During recent years several very selective Michael acceptors have been successfully

introduced in the clinic including blockbuster drugs such as ibrutinib, afatinib, or neratinib, all binding covalently to non-activated Cys-residues' – this statement of fact requires a reference. Chemical synthesis: Some of the elemental analyses are unacceptably removed from the expected values.

Reviewer #2 (Remarks to the Author):

Abdul Aziz et al report the discovery of a small molecule inhibitor that prevents the formation of cytolytic pores by the bacterial toxin pneumolysin (PLY). The inhibitors were found by virtual screening of cavities in a model of the PLY pre-pore. The rationale was that small molecules filling these cavities would prevent PLY pore formation and thus protect cells against the cytotoxic effect of PLY.

The paper shows that this approach did indeed work. Based on an initial hit identified by molecular modelling and computer screening, the authors arrived at a small molecule that inhibits pore formation. By chemical synthesis they were able to develop the inhibitor scaffold further to arrive at a more effective inhibitor. They show by haemolytic assays and electron cryo-tomography of liposomes with or without added PLY that the compound prevents pore formation. They also show that it protects human lung epithelial cells in tissue culture against lysis and cell death. Moreover, they were able to use the cavity in the PLY pre-pore in experiments as a catalyst for the assembly of the inhibitor from two precursor compounds.

The results of this study are satisfying, but the manuscript raises some serious questions.

1. Unfortunately, the limited stability of the inhibitor PB-3 prevents its practical use in therapy. Further rounds of improvements would be necessary to turn the lead compound into a useful drug against bacterial infection. It would be interesting to know whether such further rounds have been successful.
2. It is not clear from the manuscript whether the approach of cavity-based virtual screening has been useful in the development of other drugs. If there are any previous examples, they should be prominently cited.
3. Citations of the rich PLY literature are inaccurate, incomplete and, in part, misleading. Reference 10 (Marshall et al, Sci Rep 2015) reports the 2.0 Å X-ray structure of the PLY monomer, but contains no data on the average PLY pore size, nor on the number of PLY monomers in the pore, neither of which can be the result of a crystallographic analysis. Accurate data on the dimension and number of monomers in the PLY pore and pre-pore are available from other, more recent studies, which should be cited instead.
4. It is astonishing that of all the available data on the structure of PLY monomers, the authors chose to use the worst X-ray structure at 2.9 Å resolution (ref 30), even though there are three other, very much better structures of the PLY monomer at 2.5 (ref 12), 2.4 (van Pee et al, 2016) and 2.0 Å resolution (ref 10). Similarly, the authors used the lowest-resolution map of the PLY pre-pore (28 Å; ref 31) for fitting the monomer structures, in preference to a much better pre-pore map at 22 Å resolution (ref. 12). These sub-optimal choices are hard to understand, as more accurate models would most likely have resulted in different and better initial hits.
5. The use of a Volta phase plate for collecting tomographic data is no longer recommended and unnecessary. Although the phase plate can increase low-resolution contrast, it obscures high-resolution detail. Have the authors attempted to determine the number of PLY monomers in the pores or pre-pores by subtomogram averaging? If not, why not?
6. The back-to-back double pores highlighted by red circles in Fig 3a are clearly an artefact and should be identified as such.

7. It is not usual (and often not possible) to solubilize membranes directly with Amphipol-35. Instead, membranes are normally solubilized with detergent, which is then exchanged against Amphipol.

8. There are no red arrows in Figure 3c.

Reviewer #3 (Remarks to the Author):

This manuscript describes the sequential development of a small molecule inhibitor of the cholesterol binding site of Pneumolysin and other CDC toxins. It would be a potentially important therapeutic to have such a family of human-compatible inhibitors. The data support the conclusions and the structure based design is solid. However, there are many known small molecule inhibitors and there is no clear indication that this new one is any more effective than others. (for example PMID 28165051)

The authors do not clearly describe the hemolysis assay. The CDCs are very sensitive to the amount of cholesterol in the assay. Thus, when hemolytic activity is measured cholesterol has to be removed to be an accurate measure. The assay for sheep RBCs typically yields LD50 for PLN at 0.2 nM. However, in this manuscript the amount used is over 10 times more (15nM) suggesting that the assay conditions have not been optimized (such as cholesterol as an inhibitor). In addition to affecting the assay, the eventual use of these inhibitors would be in human serum where cholesterol is abundant. It would be important to test these molecules in the presence of serum.

There are many spelling and grammatical errors such as misspelling cysteine as cystein and the Strep CDC should have an A in it: intermediAlysin.

Dear reviewers,

thank you very much for reviewing our submission to Nature Communications and for inviting us to submit a revised version of our manuscript. We have modified the paper according to the suggestions of the three reviewer reports. Especially, we have included the results of the additional experiments and investigations which were proposed by the reviewers and submit a version with all changes indicated in yellow. Please note that some additional changes of the manuscript were conducted in response to the requests of the editorial team.

In the following we would like to reply to all comments and remarks of the reviewers point-by-point:

REVIEWER COMMENTS

Reviewer #1 (Remarks to the Author):

General comments:

This paper describes the design and development of small molecule inhibitors of pneumolysin (PLY), a protein from *Streptococcus pneumoniae*. PLY is a pore-forming toxin that is cytotoxic to host cells and further contributes to pathogenesis by enhancing bacterial colonization throughout the host. The aim of this study was to identify a small molecule that possessed the ability to neutralize the activity of PLY by blocking pore formation and might therefore be a potential candidate for further development as a human therapeutic.

The paper begins with a model derived from the crystallographic data of PLY monomers fitted into the cryo-EM map of the pre-pore. This enabled the identification of two putative binding pockets, one in the oligomerisation domain and one in the cholesterol binding domain. The former was chosen for this study due to its proximity to a salt bridge known to be crucial for (pre)-pore formation together with its location in an area of large-scale conformational change upon (pre)-pore formation. In silico docking of 555,000 compounds and refinement based on physicochemical properties and a pharmacophore model were carried out, giving ten scaffolds suitable for testing in an in vitro test of haemoglobin release from sheep erythrocytes. This resulted in the identification of the PB-1 series of PLY inhibitors. Two new generations of inhibitors, PB-2 and PB-3 were designed and synthesised and shown to prevent PLY-mediated damage of A549 (human lung) cells. Target engagement with PLY was shown by both bio-layer interferometry using PLY on a sensor surface, and inhibition of PLY pore formation in liposomes visualised by cryo-TEM and cryo-ET.

The binding site was then investigated further through mutational analysis of both putative pockets where it was found that the inhibitors bound not within the oligomerisation domain as expected, but rather the cholesterol binding domain. It was postulated that this was a covalent interaction due to the presence of a Michael acceptor (MA) in the hit compounds. This was confirmed in multiple ways; inactivity of an analogue without a MA, the inhibitors were unable to block PLY activity when the target cysteine was deleted, BLI showed a long residence time for PB-3 on the WT PLY and no association with the deletion mutant, and protein mass spectrometry confirmed the mass of PLY plus the ligand.

Protein templated formation of PB-3 from its constituent parts was demonstrated and specificity of the ligand investigated with a panel of MAs in the haemolysis assay where PB-3 appeared to have selectivity for PLY over other compounds in the panel. The selectivity of PB-3 for PLY over SARS-CoV2 and PTP1B (a human phosphatase), chosen as they contained activated nucleophilic cysteines, was also shown. The activity of PB-3 towards other CDCs was investigated and it was found to be a potent inhibitor of perfringolysin which contains a cysteine in the UDP sequence, but not intermedilysin, which lacks a cysteine in this position, further supporting a covalent interaction between the MA and this nucleophilic residue. Finally, it was shown using confocal laser scanning microscopy that PB-3 protected human lung cells from WT PLY during *Streptococcus pneumoniae* infection, and also from a capsid-deleted mutant where host cell damage is predominantly mediated by PLY – 100% apoptosis was observed in the absence of inhibitor, treatment with PB-3 resulted in virtually all cells surviving.

We would like to thank reviewer 1 for the precise description and evaluation of our work.

I would describe this paper as a mixed bag. The initial aim was to target inhibitors to the oligomerization domain as there was solid rationale for interfering with the pore-forming process. The in silico screening and subsequent refinement was based on this hypothesis. The hit list was filtered

based on physicochemical properties (a standard approach), then described as being re-scored based on 'frequent interactions' – it is not clear from the manuscript what this actually means, nor is a reference given.

For more clarity, we re-phrased the sentence to "The hit list was filtered by physicochemical properties and the hit molecules were re-scored by geometric fit to a 3D pharmacophore using LigandScout's geometric scoring function." Ref. 33 in the reference list describes the protocols used.

Ten compounds were selected for in vitro testing, resulting in the identification of PB-1. A small number of structural analogues were then designed and prepared, however it is not described why these particular compounds were targeted.

The first ten compounds, virtual hits VH1-10 (Supplementary Figure 3), were selected for in-vitro testing based on the virtual screening of 550,000 molecules and led to the identification of **PB-1**. Six structural derivatives of **PB-1** (shown in Supplementary Figure 5) were selected on the basis of structural similarity. In each of the tested compounds one substructural motif was changed while others were kept constant in order figure out which part of the molecule contributed to activity, and which changes improved potency. This approach delivered the more potent inhibitor **PB-2**. Next, to optimize **PB-2** for more sophisticated bacterial infection assays, the same strategy was pursued, and we synthesized 15 derivatives of **PB-2** (Supplementary Figure 7), resulting in the discovery of **PB-3**, in which the introduction of a central furane ring led to higher activities in the in vitro, cellular, and bacterial infection assays. In addition, 13 derivatives of **PB-3** were synthesized in order to scrutinize which substructural elements were essential for activity (Supplementary Figure 8).

Molecular dynamic simulations were done to identify a binding mode and interactions, (for some reason figure 1d only appears in the supplementary, not the manuscript) but this only showed interactions that could easily be identified by an experienced medical chemist. There is no real rationale given for the changes made, and very little discernable structure-activity relationship apart from the requirement for a MA. Additionally, some compounds in the experimental are not shown in Table 1.

Yes, the reviewer is correct: in Figure 1d we show the structure of the primary screening hit. Supplementary Figure 1d adds to this the interaction frequencies determined by MD simulation. We have adjusted the figure captions correspondingly to make these differences clearer. We have limited the structures shown in the paper in Table 1 to those few which indicate the requirement for the Michael acceptor. More detailed structure-activity relationships are shown in the SI part analyzing the contributions of other structural elements to the activity of **PB1-3** in Supplementary Figures by reporting the activity data for 56 derivatives of PBs. We have discussed these data in the text, however, find the amount of information too detailed to show all data in the paper itself.

I note here that a series of small molecules containing a very similar MA that inhibit the pore-forming protein perforin have been published previously but are not referenced here (Lena et al 2008, DOI 10.1021/jm801063n).

Thank you for drawing this interesting link to this human pore-forming protein perforin which we have added to our reference list as ref.# 41. Please note, that the small molecules reported in this paper do not contain a reactive Michael acceptor and are not described as covalent inhibitors of perforin. The perforin inhibitors do contain an exocyclic double bond, however, this double bond is not activated by conjugation with neighboring electron-withdrawing groups such as carbonyl or cyano residues.

The instability of PB-1 and 2 is reported, but how the more stable PB-3 is arrived at is not explicitly described, clearly more compounds were made than just those reported here.

PB-3 is active in the cellular infection assays for about 4 h. Yes, altogether 56 compounds were made for identification and SAR of **PB-3**. We have added these requested details to the text.

The PB compounds were then shown to prevent PLY induced damage in human lung cells and target engagement with PLY shown as described above. The actual binding site was then identified and characterised through an elegant series of mutagenesis experiments. Although the PB compounds appear to show selectivity (when compared against SARS-CoV2, PTP1B and a non-cysteine containing CDC), the compounds were not tested against human pore-forming proteins such as perforin. This would be a key experiment due to the possibility of off-target effects involving the

immune system.

In summary, there is a disconnect between the initial aim and the actual outcome in this paper. Although a well-argued rationale was given for the target pocket and compound design, the study was essentially a very small in vitro screen of compounds that resulted in a modest inhibitor of PLY activity. Cellular experiments were carried out at a high concentration of inhibitor; 6 μM , single or multiple doses – the IC_{50} is 3.1 μM - it is not clear how test concentrations were determined?

Cellular experiments were conducted at varying concentrations starting from 1 μM and reaching about 20% inhibition of PLY. Concentrations were successively raised to 6 μM reaching >90% inhibition of PLY after 24 h.

Inhibitors should also have been tested for off-target effects against CTL and/or NK cells to check if inhibition of the immune protein perforin, which possesses a similar pore-forming mechanism of action to the CDCs, affected lytic activity. The lipophilicity of the hit compound (PB-3 $\text{cLogP} = 6.56$) could be a significant driver of binding, leading to a host of off-target effects. Although the physicochemical properties are not reported, these compounds are unlikely to be very soluble and don't look much like 'drug-like' leads. While there are many very interesting and elegant experiments carried out to characterise these inhibitors, if the *raison d'être* of this paper is to identify 'lead structures for the preclinical development of PLY inhibitors' for eventual clinical use, then we are a long way from that point. Another approach to take with this story would be to relocate the medicinal chemistry/compound design into a separate paper (and add in any compounds that don't appear here) and in this paper focus on the current work where PB-2/3 were used as chemical tools to elucidate the mechanism of action for this scaffold.

To evaluate the interaction of **PB-3** with human perforin, we teamed up with colleagues from the University Medical Center and the Leibniz Institute of Virology in Hamburg and conducted a functional NK-cell cytotoxicity assay using the perforin-sensitive K562 target cell line (Lehmann *et al.*, 2000, PMID: 10887123).

The predominant way of NK cell killing is by release of lytic granules that contain granzymes and perforin. This process called degranulation can be measured by translocation of lysosomal-associated membrane protein-1 (LAMP-1/CD107a) to the cell surface of the NK cell, reduction of intracellular perforin content, and ultimately amount of target cell lysis (Gwalani and Orange, 2018, <https://doi.org/10.4049/jimmunol.1701500>).

In our experimental setup, K562 cells were co-cultured with primary rested NK cells of three individual donors at an effector: target ratio of 2:1 in the presence of varying concentrations (1 μM , 3 μM , and 6 μM) of the compound **PB-3**, 10 mM EDTA (as positive control for inhibition of lysis) or the vehicle control (DMSO) for 3 hours (Supplementary Figure 16a & b). Per donor, two technical replicates for lysis were performed and mean values are depicted (Supplementary Figure 16b). First, we tested that the compound (at 6 μM) was not cytotoxic for K562 cells alone (Supplementary Figure 16b). No inhibition of NK cell cytotoxicity was observed with addition of 1 μM and 3 μM of the compound. At 6 μM there was a minimal reduction in K562 lysis that was accompanied by a decrease in release of lytic granules (as measured by CD107a surface expression) indicating that **PB-3** is not an inhibitor of human perforin at concentrations that fully inhibited PLY and other CDC tested in this paper.

More specific observations:

P3: Reference 26 appears before 19-25

We have fixed this.

P4: The statement 'however, these molecules have been reported active toward numerous proteins and against many diseases without adequate biological or clinical evidence' – is this the opinion of the authors or the reference given (26)? This statement needs further substantiation.

This statement indeed is a summary of the cited paper 26 (now 27), which gives a general overview of the virtually countless reported bioactivities of polyphenolic compounds observed in many biological assays and with many proteins. At the same time, none of these compounds have been clinically admitted for therapeutic use.

To substantiate this statement, we have given in the preceding sentence several examples for unspecific or poly-specific protein binders which have been reported to interfere with the activity of PLY. These include the natural flavonoids epigallocatechin (ref. 19, PMID: 15019969), amentoflavone

(ref. 20, PMID: 22767439), apigenin (ref. 21, PMID: 10751547), and quercetin (ref. 22, PMID: 15019969), or other polyphenols such as juglone (ref. 24, PMID: 35656011), verbascoside (ref. 25, PMID: 1812212), or shikonin (ref. 26, PMID: 31553936). All these polyphenolic compounds were found to interact with many other proteins as well and to interfere with numerous biological functions without a defined binding site or binding interaction.

The polyphenol quercetin appears to be an especially good example to illustrate this. In 2021, a literature analysis report (Animal Science Papers and Reports vol. 39 (2021) no. 3, 199-212) counted 15,865 papers on quercetin, most of them published since 2000. Medical use of quercetin was proposed in 890 papers for almost every indication from cancer, Alzheimer, Parkinson, obesity, hypertension, diabetes, and rheumatoid arthritis to corona infections. Many protein targets and cellular mechanisms are discussed as well as general redox effects.

To make the value of our new inhibitors clearer to the readers and to distinguish them from the old, unspecific inhibitors, we are introducing them now as "first targeted inhibitors of PLY" in the revised version of the paper.

P5: How many compounds were tested in total? They don't all appear to be here.

All together we have tested 56 molecules for this study including the primary in vitro testing, all synthetic variations and similar Michael acceptors for specificity testing. We have added all compounds to the supplementary part (Supplementary Figures. 3, 5, 7, 8, and 14) to provide the more detailed SAR requested by this reviewer.

P5: Fig 1d is referred to here, but is in the supplementary, not the main text.

Thanks for the hint, we have fixed this issue.

Table 1: Compounds 2.6, 2.8, 2.10 are shown in the experimental but do not appear in the table with data.

Compounds 2.7, 2.9, 2.11 (if they exist) do not appear at all.

The complete list of tested derivatives of **PB-2** is reported in Supplementary Figure 7. For Table 1 in the paper, we have selected only those molecules whose activities contribute to a deeper understanding of the structure-activity relationships of these compounds, namely, **PB-2.6, 2.7, 2.8, 2.9, 2.10, and 2.11.**

P8: What compound concentrations were used in the 'PLY-inhibition protects human lung cells' A549 assay?

The LDH assays were conducted at concentrations of 1, 3, and 6 μM of **PB-3** as shown in Figure 2d and e; strongly significant protection of A549 cells was observed at concentrations of 3 and 6 μM with $p=0.0009$ and $p=0.0002$ for 4 and 24 h, respectively. For higher clarity, we have added this information also to the manuscript text. Same highest concentration (6 μM) was used for the microscopy assays. Concentrations for **PB-1** and **PB-2** are reported in Supplementary Figure 8, as stated.

P8: 'Cellular lysis induced by PLY was quantified by determining the activity of lactate dehydrogenase (LDH) in each sample' Why this assay? The assay choice needs further explanation.

The LDH assay was selected as it provides direct evidence for pore formation resulting in the release of the enzyme LDH from the cytosol of the cells. This is quite different from other cytotoxicity assays such as the MTT or Alamar blue assays which merely quantify the overall amounts of cellular reductants without reporting on the lysis of the cellular membrane. We have added a statement of this to the manuscript text and to the method section to explain the assay selection.

P8: Ref 38 appears before 37.

Thanks for pointing this out, we fixed it.

P11: It is not clear why the pore-blocking assay using cryo-TEM was carried out using PB-2; wouldn't it be better to use the more stable PB-3 as is done elsewhere?

The cryo-TEM experiments with **PB-2** were conducted before **PB-3** became available. Considering the costs of these experiments, we did not repeat them with the more potent **PB-3**, however, in the light of all other data received for **PB-3** there is no doubt that the improved inhibitor **PB-3** would also inhibit pore formation in this assay.

P11: Refers to figure 3d, there is no 3d (means 3a?)

It is indeed Figure 3a. We fixed this and thank you for pointing this out.

P13: The findings support or suggest, rather than establish binding to Cys428.

We are now using the term “suggested.”

P14: ‘During recent years several very selective Michael acceptors have been successfully introduced in the clinic including blockbuster drugs such as ibrutinib, afatinib, or neratinib, all binding covalently to non-activated Cys-residues’ – this statement of fact requires a reference.

We have provided the reference for this statement as ref. 39 in our list.

Chemical synthesis: Some of the elemental analyses are unacceptably removed from the expected values.

The elemental analysis of **PB-2.3** was repeated as requested. It is now correct and has been added to the method section.

Reviewer #2 (Remarks to the Author):

Abdul Aziz et al report the discovery of a small molecule inhibitor that prevents the formation of cytolytic pores by the bacterial toxin pneumolysin (PLY). The inhibitors were found by virtual screening of cavities in a model of the PLY pre-pore. The rationale was that small molecules filling these cavities would prevent PLY pore formation and thus protect cells against the cytotoxic effect of PLY.

The paper shows that this approach did indeed work. Based on an initial hit identified by molecular modelling and computer screening, the authors arrived at a small molecule that inhibits pore formation. By chemical synthesis they were able to develop the inhibitor scaffold further to arrive at a more effective inhibitor. They show by haemolytic assays and electron cryo-tomography of liposomes with or without added PLY that the compound prevents pore formation. They also show that it protects human lung epithelial cells in tissue culture against lysis and cell death. Moreover, they were able to use the cavity in the PLY pre-pore in experiments as a catalyst for the assembly of the inhibitor from two precursor compounds.

The results of this study are satisfying, but the manuscript raises some serious questions.

1. Unfortunately, the limited stability of the inhibitor PB-3 prevents its practical use in therapy. Further rounds of improvements would be necessary to turn the lead compound into a useful drug against bacterial infection. It would be interesting to know whether such further rounds have been successful.

We synthesized 13 derivatives of **PB-3** and now added to supplementary data (Supplementary Figure 8).

2. It is not clear from the manuscript whether the approach of cavity-based virtual screening has been useful in the development of other drugs. If there are any previous examples, they should be prominently cited.

Structure-based virtual screening using 3D pharmacophores is an approach, which has been widely used for the development of drug leads, by us and by other groups. We have added a sentence stating this to the paragraph on structure-based screening in the results section and cite a reference describing this method and reviewing prominent applications of it (ref. 33). We are further citing three additional references describing the algorithm used for flexible docking (ref. 48), the application of molecular dynamics for evaluating the binding mode (ref. 49) and the application of this methodology in the screening of GPCR as targets (ref. 50).

3. Citations of the rich PLY literature are inaccurate, incomplete and, in part, misleading. Reference 10 (Marshall et al, Sci Rep 2015) reports the 2.0 Å X-ray structure of the PLY monomer, but contains no data on the average PLY pore size, nor on the number of PLY monomers in the pore, neither of which can be the result of a crystallographic analysis. Accurate data on the dimension and number of monomers in the PLY pore and pre-pore are available from other, more recent studies, which should be cited instead.

Thank you for pointing that out. The reference was corrected. It is now ref. 12.

4. It is astonishing that of all the available data on the structure of PLY monomers, the authors chose to use the worst X-ray structure at 2.9 Å resolution (ref 30), even though there are three other, very much better structures of the PLY monomer at 2.5 (ref 12), 2.4 (van Pee et al, 2016) and 2.0 Å resolution (ref 10). Similarly, the authors used the lowest-resolution map of the PLY pre-pore (28 Å; ref 31) for fitting the monomer structures, in preference to a much better pre-pore map at 22 Å resolution (ref. 12). These sub-optimal choices are hard to understand, as more accurate models would most likely have resulted in different and better initial hits.

We started the project with PDB entry 4ZGH (Ref 30) because it was the best crystal structure available at the time. With the subsequent release of other crystal structures, e.g. PDB entry 5AOD, we checked for potential new structural insights and consistency with our model. Due to the high similarity, we decided to continue with the model based on 4ZGH. To highlight the high structural similarity, we have generated an overlay of all published structures shown in Supporting Figure 1 below. Despite the better overall resolution, we didn't include the crystal structure with pdb entry 5CR6 in our study because it contains a mutation in the cholesterol binding region (C428A). This region of the protein is essential for the function of PLY and the specific cysteine residue (C428) is the one that reacts with **PB-3**. Regarding the pore map, the PLY pre-pore used in this manuscript (ref 31) is the only one available with coordinates allowing for direct atom-wise mapping of the crystal structure to the alpha trace.

	1	2	3	4	5
1:4ZGH.A		1.48	1.54	1.57	1.46
2:6JMP.A	1.48		1.52	1.50	1.48
3:5AOE.A	1.37	1.51		0.55	0.91
4:5AOE.B	1.38	1.49	0.51		0.97
5:5AOF.A	1.30	1.48	0.85	0.89	

Supporting Figure 1: Overlay of PLY crystal structure 4ZGH (blue) aligned with 6JMP, 5AOE and 5AOF (gray) showing high similarity (left). RMSD similarity matrix in Å (right).

5. The use of a Volta phase plate for collecting tomographic data is no longer recommended and unnecessary. Although the phase plate can increase low-resolution contrast, it obscures high-resolution detail. Have the authors attempted to determine the number of PLY monomers in the pores or pre-pores by subtomogram averaging? If not, why not?

We have used the Volta phase plate to acquire close-to-focus tilt series where phase reversal correction is unnecessary and easy alignment is possible due to enhanced contrast. We agree with the reviewer that the phase plate emphasizes low frequency characteristics in the raw data, but the effect can partially be overcome by band-pass filtering.

We have not tried to determine the number of PLY monomers of the pore or pre-pore. This would have required a different approach with corresponding larger data sets. However, the detailed structural elucidation of the pores and pre-pores has already been published (van Pee et al., 2017, ref. 12) and was therefore not the subject of our investigation.

Aim of our tomographic study was to demonstrate a) that our recombinant PLY interacts effectively with cholesterol-containing liposomes and to form pores, and b) that our inhibitors effectively block this interaction. Both aims have been reached.

6. The back-to-back double pores highlighted by red circles in Fig 3a are clearly an artefact and should be identified as such.

Pre-pore and pore formation of PLY in cholesterol-containing liposomes have been reported by other researchers as well (see ref. 38) and seem to be the expected outcome of the reported experiments. As the back-to-back prepore or pore in Figure 3a might have appeared to be ambiguous, we have replaced Figure 3a with another 3D picture from the same experiment where the interaction of PLY with the membrane as well as pore and pre-pore formation can be more clearly observed.

7. It is not usual (and often not possible) to solubilize membranes directly with Amphipol-35. Instead, membranes are normally solubilized with detergent, which is then exchanged against Amphipol.

This is exactly true. As reported in our methods part we have solubilized the liposome membranes with CYMAL-6 overnight first and subsequently Amphipol was added to replace the detergent. For clarity we have added this detail also to the main text.

8. There are no red arrows in Figure 3c.

Thanks for the hint. We had only added the arrows in the videos, yet, and have added them now in Figure 3c as well.

Reviewer #3 (Remarks to the Author):

This manuscript describes the sequential development of a small molecule inhibitor of the cholesterol binding site of Pneumolysin and other CDC toxins. It would be a potentially important therapeutic to have such a family of human-compatible inhibitors. The data support the conclusions and the structure based design is solid. However, there are many known small molecule inhibitors and there is no clear indication that this new one is any more effective than others. (for example PMID 28165051)

There are no reported drug-like small molecule inhibitors interacting specifically with PLY, yet. The paper mentioned by reviewer #3 (cited as ref. 27 in our manuscript) describes the use of cholesterol, sitosterol, and other natural steroids. As these steroids are insoluble in aqueous buffers, they can only be used in the form of liposomal preparations embedded into membranes. It has been reported (ref. 29) that cholesterol-containing liposomes are able to scavenge cholesterol-binding proteins such as PLY, however, such liposomes cannot be considered as drugs, especially as loading the organism with cholesterol induces severe health risks including atherosclerosis, cardiac infarction, and ischemic stroke. As discussed above, all other reported "PLY inhibitors" are non-specific or poly-specific protein binders, molecules which interact with many different proteins and have countless biological effects such as the flavonoids quercetin, apigenin, amentoflavone, verbascoside, epigallocatechin, etc. A specific molecular interaction of these molecules with PLY or other CDC has not been reported, yet.

The authors do not clearly describe the hemolysis assay. The CDCs are very sensitive to the amount of cholesterol in the assay. Thus, when hemolytic activity is measured cholesterol has to be removed to be an accurate measure. The assay for sheep RBCs typically yields LD50 for PLN at 0.2 nM. However, in this manuscript the amount used is over 10 times more (15 nM) suggesting that the assay conditions have not been optimized (such as cholesterol as an inhibitor). In addition to affecting the assay, the eventual use of these inhibitors would be in human serum where cholesterol is abundant. It would be important to test these molecules in the presence of serum.

We have carefully optimized the hemolysis assay before conducting this study. Sheep erythrocytes were isolated by centrifugation and obtained cholesterol-free by washing three times with PBS prior to use in the hemolysis assay. In the hemolysis assay, the LD_{50} of PLY was 0.5 nM, nearly identical to the value reported in the literature (see Supplementary Figure 4a). At the PLY concentration of the LD_{50} , however, only 50% of the erythrocytes are lysed. For conducting a sensitive and robust assay and for avoiding false positives in the screening, we raised the PLY concentration to 15 nM resulting in >95% hemolysis.

Following the suggestion of this reviewer, we have repeated the hemolysis assay in presence of 2.5% (v/v) human serum (Supplementary Figure 4b). As expected, addition of human serum increases the LD_{50} of PLY from 0.5 to 2.6 nM, which can be rationalized by the abundance of PLY-deactivating cholesterol in human serum. Addition of serum to the hemolysis assay amplifies the inhibitory potential of our PLY-inhibitors (Supplementary Figure 4c). The IC_{50} value of **PB-3**, for example, is reduced from 3.1 μ M in the serum-free assay to 0.22 μ M in the serum-containing assay. The observed activation of our PLY inhibitors by addition of serum can be explained by the deactivation of PLY through serum cholesterol as well. We have added a statement on the hemolysis assay conducted with serum to the manuscript text and the IC_{50} value to Table 1. Moreover, conducting the hemolysis assay with added serum suggests that the washed erythrocytes and the recombinant PLY were indeed cholesterol-free.

There are many spelling and grammatical errors such as misspelling cysteine as cystein and the Strep CDC should have an A in it: intermediAlysin.

Thanks for these orthographic corrections. We have checked the internet for the prevalent spelling of ILY and found intermediylisin as the dominant form with about 9100 hits, while in less than 80 cases the term intermedialsin was used. Thus, we would like to stick to the spelling used in the first submission of this paper.

In summary, we have conducted all requested additional experiments, and added the results to the manuscript. We hope to have answered all questions of the reviewers and the editors to their full satisfaction and have revised the paper accordingly.

Thus, we are looking forward to the decision on our paper,

sincerely,

Jörg Rademann

REVIEWER COMMENTS

Reviewer #1 (Remarks to the Author):

Thank you for submitting a revised version of this manuscript. The modified explanations of the medicinal chemistry have clarified the evolution of the lead compounds PB-1 to PB-3 considerably, and link nicely to the data in the supporting information. The SI data is well organised in a logical fashion and referenced appropriately in the main text. Additional explanations around some of the other aspects of the work have also enhanced the readability of this manuscript considerably. I would also like to thank the authors for doing the additional work to show the lead compound does not block NK-killing as if this activity was observed it would be highly problematic in a human therapeutic targeting the toxic effects of PLY.

Finally, to address the author's addition of ref 41 and observation that these compounds were not Michael acceptors - this is quite correct, and my error. The reference I was actually referring to was in fact Bioorganic and Medicinal Chemistry, 20 (3), 1319-1336, 2012. (<https://doi.org/10.1016/j.bmc.2011.12.011>) These inhibitors of perforin are Michael acceptors, exist as interconverting mixtures of E and Z isomers, and are very similar in structure to the compounds of the current paper. Note that this problem was resolved in a later publication; Bioorganic and Medicinal Chemistry Letters, 27, 1050-1054, 2017. (<https://doi.org/10.1016/j.bmcl.2016.12.057>). My apologies for the confusion!

Overall, I found the current version of the manuscript to be a much more coherent description of an interesting piece of work with potentially important applications and would therefore now recommend it for publication.

Reviewer #2 (Remarks to the Author):

The authors have revised their manuscript thoroughly in response to the very thorough reviewer comments. It is now almost ready for being accepted, with the exception of two small but important details:

1. In response to comment 1 of reviewer 2, the authors have added 13 more derivatives of PB-3 to the manuscript. However, no information is provided as to whether these new derivatives are any more stable or more useful than PB-3, which was the point of the question.
2. Line 934 of the revised manuscript, PB-2.3 should be replaced by PB-3

Reviewer #3 (Remarks to the Author):

The authors have undertaken a comprehensive response to reviewer 3 comments. The medical relevance of a PLN inhibitor is clear and its use would be clinically impactful. The new data provided indicates the inhibitor is effective in serum, a finding that is required to support eventual clinical utility. The detailed questions of methods from Reviewers 1 and 2 indicate significant points needing clarification to support novelty and rigor of the discovery methods.

Dear reviewers,

thanks for your additional comments and for the invitation to submit a second revision, in which we have addressed all open points. In the following, we are answering all of your open questions point-by-point:

REVIEWER COMMENTS

Reviewer #1 (Remarks to the Author):

Thank you for submitting a revised version of this manuscript. The modified explanations of the medicinal chemistry have clarified the evolution of the lead compounds PB-1 to PB-3 considerably, and link nicely to the data in the supporting information. The SI data is well organised in a logical fashion and referenced appropriately in the main text. Additional explanations around some of the other aspects of the work have also enhanced the readability of this manuscript considerably. I would also like to thank the authors for doing the additional work to show the lead compound does not block NK-killing as if this activity was observed it would be highly problematic in a human therapeutic targeting the toxic effects of PLY.

Finally, to address the author's addition of ref 41 and observation that these compounds were not Michael acceptors - this is quite correct, and my error. The reference I was actually referring to was in fact Bioorganic and Medicinal Chemistry, 20 (3), 1319-1336, 2012. (<https://doi.org/10.1016/j.bmc.2011.12.011>) These inhibitors of perforin are Michael acceptors, exist as interconverting mixtures of E and Z isomers, and are very similar in structure to the compounds of the current paper. Note that this problem was resolved in a later publication; Bioorganic and Medicinal Chemistry Letters, 27, 1050-1054, 2017. (<https://doi.org/10.1016/j.bmcl.2016.12.057>). My apologies for the confusion!

Overall, I found the current version of the manuscript to be a much more coherent description of an interesting piece of work with potentially important applications and would therefore now recommend it for publication.

Thank you very much for this positive evaluation of our revision and for the recommendation to publish our article.

Reviewer #2 (Remarks to the Author):

The authors have revised their manuscript thoroughly in response to the very thorough reviewer comments. It is now almost ready for being accepted,

Thank you for acknowledging the thoroughness of our revision and for denominating it as "almost ready".

with the exception of two small but important details:

1. In response to comment 1 of reviewer 2, the authors have added 13 more derivatives of PB-3 to the manuscript. However, no information is provided as to whether these new derivatives are any more stable or more useful than PB-3, which was the point of the question.

One important goal in the optimization of **PB-2** to **PB-3** was to increase the chemical (hydrolytic) stability of these PLY inhibitors. As a result, **PB-3** showed only very little decomposition in PBS buffer at pH 7.4 and 37 °C as compared to **PB-2**. To quantify hydrolysis of PB-compounds, we have employed an HPLC method. The isosbestic point (wavelength of identical absorption coefficient of two compounds) of **PB-3** and its decomposition product, aldehyde **PB-3.2**, was determined by UV-vis spectroscopy at 356 nm, and the HPLC detector was set to this wavelength in order to enable the quantitative determination of starting material and hydrolysis product. While **PB-2** and several derivatives of **PB-3** showed rapid hydrolysis, hydrolysis of **PB-3** proved to be very slow with only about 4.8% decomposition after 6 h in PBS (pH 7.4). We have now added the HPLC runs of **PB-3** showing compound decomposition as Supplementary Figures 8b to make these differences clear to the reviewers and to the public/ readership of the journal. For comparison we also show the HPLC chromatograms indicating the rapid hydrolysis of two derivatives of **PB-3**, **PB-3.12** and **PB-3.13** in Supplementary Figures 8c and 8d. We have also added a reference to these stability studies in the manuscript stating that "... in **PB-3** increased the chemical stability of the PLY inhibitor considerably with less than 5% hydrolysis after 6 h in PBS (pH 7.4) at 37 °C (see Supplementary Figure 2b-d)".

Compared to the Knoevenagel products of barbituric acid (like **PB-3**, **PB-3.12** and **3.13**), cyano esters and ketones (like **PB-3.3-7**) were completely stable. Their reduced activity thus can be contributed to the reduced reactivity with the Cys-residue of PLY.

2. Line 934 of the revised manuscript, PB-2.3 should be replaced by PB-3

We have corrected this error, thanks for finding.

Reviewer #3 (Remarks to the Author):

The authors have undertaken a comprehensive response to reviewer 3 comments. The medical relevance of a PLN inhibitor is clear and its use would be clinically impactful. The new data provided indicates the inhibitor is effective in serum, a finding that is required to support eventual clinical utility.

We are happy to hear that reviewer 3 acknowledges the comprehensive response to his/her comments and emphasizes the potential clinical impact and utility of our results.

The detailed questions of methods from Reviewers 1 and 2 indicate significant points needing clarification to support novelty and rigor of the discovery methods.

Yes, we agree with reviewer 3 that reviewers 1 and 2 have raised significant points and we are happy that reviewers 1 and 2 have been satisfied with our clarifications.

With this, we have answered the questions of the reviewers to our best and hope for a positive evaluation,

yours sincerely,

Jörg Rademann

REVIEWER COMMENTS

Reviewer #4 (Remarks to the Author):

In the manuscript "A Targeted Small Molecule Inhibitor with Elongated Residence Time Blocking the Cytolytic Effects of Pneumolysin and Homologous Toxins" the authors have set off to unravel possible new small molecule inhibitors of the toxin pneumolysin. Structural screening has resulted in the potential inhibitor PB-1, substructural variations of PB-1 resulted in PB-2 and systematic variation of the substituents of the reactive double bond led to discovery of the most potent compound PB-3. The latest is suggested to covalently, yet reversibly, bind to Cys428.

I find this article and the work inside absolutely essential for further development of novel inhibitors against PLY, and Cys-containing toxins in general.

However, the work still requires significant improvements of its consistency. For example, even though the most active and studied compound is PB-3, the cryo-TEM investigations of PBs' impact on liposomes is done only for PB-2, even though the title of the legend in Figure 3 claims "Analysis of PB-inhibitors via Cryo-TEM". Definitely, PB-3's effects on the liposomes should be evaluated as well. The same is true for molecular dynamics simulations of the binding of the substances to PLY. Only PB-1 is simulated, whose mode of binding is not discussed in the article at all. The binding modes from docking of all three substances should be systematically compared (PB-3 docking mode is shown from some kind of a side view+zoom in in Figure 4), potential binding site of ??? is shown in Supplementary Figure 1b and the residues around PB-1 and PB-2 are schematically shown in Supplementary Figure 1e. The visualisation of these three poses should be unified in order to see the similarities and differences. Moreover, the current statement in the discussion "... suggested a covalent reversible mode-of-action of PB-3 as PLY inhibitor. This mode-of-action was supported by a plausible structure of the covalent inhibitor-protein complex derived from covalent docking and molecular dynamics simulation." is currently not supported by any data, as MD simulations have been performed for PB-1 only. Thus, molecular dynamics simulations should be performed for all three substances PB-1, PB-2, PB-3, stating which force field was used and whether PLY was monomeric or dimeric. Moreover, the covalent bond between PB-3 and PLY should be included in an additional set of simulations in order to estimate the role of PB-3 covalent binding on the structure of the PLY monomer and its interaction with (potentially) neighboring PLY. Moreover, comprehensive analysis of the simulations should be performed, described and included at least in the supplementary. Alternatively, all current MD simulations and MD-simulation statements shall be removed from the manuscript in order to avoid misleading of the scientific community by drawing general conclusions for all three PB-1, PB-2 and PB-3 based on PB-1 only, whose mode of binding could be different (it is impossible to judge based on the current state of the description of the results and of the methods. Based on Suppl. Fig 1e, neither PB-1 nor PB-2 are in contact with Cys428. Also it is completely unclear which interaction frequencies are shown in Suppl. Fig 1d. Interactions with PLY in general? Or specific interactions which were detected by docking?)

Apart from these major concerns I suggest to correct/improve following minor issues:

Lines 157-158: "PB-3 was the most active PLY inhibitor of 13 structural derivatives, PB-3.1-PB-3.13" here PB-3 equals PB3-1 as it is missing in the Supplementary figure 8, but this is not stated anywhere. Writing something like "PB-3 was significantly more active than any of its 12 derivatives PB-3.2-PB-3.13." would increase the understandability.

Figure 2a. I have printed the pdf on our HPC laser printer and can not distinguish the colors for PB-1 and PB-2. using a different color, e.g. for PB-1 would be of a great help.

Figure 2. in the legend subfigure g is mentioned, but not subfigure g is present

line 266 typo: "inactive hemolysis"

line 290 corrupted sentence (and suggested a binding mode in which PB-3 through hydrophobic

interactions with the amino acid residues of domain 4 with (Figure 4f)."

lines 306-318: yellow section on perforin doesn't fit to the flow of this paragraph, I would move it to the next paragraph as an additional independent control. or make a new paragraph...

line 347 "All three PB-compounds did not inhibit ..." -> "None of the three PB-compounds inhibited..."

lines 349-357 speak mostly about PB-3, therefore I would move the statement "The protective effect of PB-2" to the end of the paragraph. PB-3... similar protective effects were also observed for PB-2 (supplementary Figure 9). otherwise the discussion of the results for PB-2 is missing in the paragraph.

line 363. "nuclear executioner caspase activation" what is this? executioner of caspase activation? The legend of Figure 4 is corrupt in multiple ways. line 375 typo "muatnt", line 378 "and unable to neutralize" -> "and is unable to neutralize", same line typo "concnetrations" line 382 "PLY (e) intermedilysin" -> "PLY. (e) Intermedilysin" ; the description of the distinct samples in f belongs to e. Legend f: " A docking suggested binding mode of PB-3." -> " A binding mode of PB-3 suggested by docking..." some more details??? you described nowhere the docking of PB-3

Discussion. The sentence " Binding of PB-2 inhibited membrane insertion of PLY and the formation of PLY pores as visualized by cryo-transmission electron tomography of cholesterol-containing liposomes and resulted in protein precipitation." is not connected to the sentence before and afterwards as it is not clear why cryo-TEM was performed for PB-2 only?

Methods: details on MD simulations like force field, protein structure (monomer? dimer?) presence of lipids? ion concentration? and details on analysis missing.

Dear reviewer,

Thank you for the careful evaluation of our work. We have modified the manuscript according to your suggestions and reply in the following to the raised questions point-by-point:

Reviewer #4 (Remarks to the Author):

In the manuscript "A Targeted Small Molecule Inhibitor with Elongated Residence Time Blocking the Cytolytic Effects of Pneumolysin and Homologous Toxins" the authors have set off to unravel possible new small molecule inhibitors of the toxin pneumolysin. Structural screening has resulted in the potential inhibitor PB-1, substructural variations of PB-1 resulted in PB-2 and systematic variation of the substituents of the reactive double bond led to discovery of the most potent compound PB-3. The latest is suggested to covalently, yet reversibly, bind to Cys428.

I find this article and the work inside absolutely essential for further development of novel inhibitors against PLY, and Cys-containing toxins in general.

Thank you very much for this positive and supportive evaluation of our article which is in agreement with the reviewer reports of the preceding three reviewers.

However, the work still requires significant improvements of its consistency. For example, even though the most active and studied compound is PB-3, the cryo-TEM investigations of PBs' impact on liposomes is done only for PB-2, even though the title of the legend in Figure 3 claims "Analysis of PB-inhibitors via Cryo-TEM". Definitely, PB-3's effects on the liposomes should be evaluated as well.

We have conducted the cryo-EM experiments to prove that the recombinant PLY used in our studies inserts into liposomal membranes containing cholesterol and forms pre-pore and pore structures within these membranes. We have also shown by cryo-EM that PB-2 effectively inhibits membrane insertion and pore formation. These extended and very cost-intensive studies were conducted with PB-2 only as at that time PB-3 had not been developed, yet.

We have, however, demonstrated in this paper the mode-of-action of both PB-2 and PB-3 in detail, studying the effective inhibition of hemolysis, the protection of lung epithelial cells, the binding kinetics (on- and off-rates of binding) using BLI, and have verified the binding site using the PLY mutants and demonstrated the covalent attachment using structural derivatives. In all these experiments PB-2 and PB-3 behaved identical, so that there is no doubt that also PB-3 is able to block pore formation effectively.

To avoid any misunderstanding and to make the caption of Figure 3 more precise, we have changed the text to "Investigation of PLY and PB-2 using Cryo-EM". As we stated in the abstract of the manuscript already before that "Cryo-electron tomography revealed that PB-2 blocks PLY-binding to cholesterol-containing membranes and subsequent pore formation", this change removes the pointed-out inconsistency.

The same is true for molecular dynamics simulations of the binding of the substances to PLY. Only PB-1 is simulated, whose mode of binding is not discussed in the article at all. The binding modes from docking of all three substances should be systematically compared (PB-3 docking mode is shown from some kind of a side view+zoom in in Figure 4), potential binding site of ??? is shown in Supplementary Figure 1b and the residues around PB-1 and PB-2 are schematically shown in Supplementary Figure 1e. The visualisation of these three poses should be unified in order to see the similarities and differences. Moreover, the current statement in the discussion "... suggested a covalent reversible mode-of-action of PB-3 as PLY inhibitor. This mode-of-action was supported by a plausible structure of the covalent inhibitor-protein complex derived from covalent docking and molecular dynamics simulation." is currently not supported by any data, as MD simulations have been performed for PB-1 only. Thus, molecular dynamics simulations should be performed for all three substances PB-1, PB-2, PB-3, stating which force field was used and whether PLY was monomeric or dimeric. Moreover, the covalent bound between PB-3 and PLY should be included in an additional set of simulations in order to estimate the role of PB-3 covalent binding on the structure of the PLY monomer and its interaction with (potentially) neighboring PLY. Moreover, comprehensive analysis of the simulations should be performed, described and included at least in the supplementary. Alternatively, all current MD simulations and MD-simulation statements shall be removed from the manuscript in order to avoid misleading of the scientific community by drawing general conclusions for all three PB-1, PB-2 and PB-3 based on PB-1 only, whose mode of binding could be different (it is impossible to judge based on the current state of the description of the results and of the methods. Based on Suppl. Fig 1e, neither PB-1 not PB-2 are in contact with Cys428. Also it is completely unclear which interaction frequencies are shown in Suppl. Fig 1d. Interactions with PLY in general? Or specific interactions which were detected by docking?)

To make the logic and the presentation of the modeling data understandable, we need to explain the proceedings and the central findings of our work.

In short:

- Generation and inspection of the protein model as described, suggested two alternative binding sites for small molecules as shown in Figure 1a,b, one in the cholesterol binding domain and a second one at the interface between two protein monomers, in the oligomerization domain. Virtual screening was conducted only for the

postulated binding pocket at the oligomerization site as this binding site was less flexible and thus seemed to be better suited for small molecule binding.

- PB-1 was identified as one of the virtual screening hits and was the only biologically active and sufficiently soluble hit. Therefore, we generated the MD simulation of PB-1 and showed it in Supplementary Figure 1. The binding site is located at the dimer interface, therefore all simulations were conducted using a protein dimer as described in the methods section. The force field used is described in the methods part. The docking shows no cysteine residue, as there is no cysteine residue in this binding site.
- subsequent chemical, biochemical, and biophysical data of PB-1, PB-2, and PB-3 disproved the original binding hypothesis of PB-1 and instead suggested covalent reversible binding to Cys428 (in the undecapeptide at the cholesterol binding site), far remote from the originally proposed oligomerization site. This binding hypothesis was further substantiated by the studies with mutants of PLY, by chemical derivatization, and by studying the effects on perfringolysin (has a cysteine and is inhibited) and intermedilysin (has no cysteine and is not inhibited).
- as these findings disproved the original binding hypothesis for PB-1, we did not conduct additional MD for PB-2 and PB-3 in the oligomerization site. The reviewer proposes to do so, however, we do not find it scientifically useful to generate further MD data for a binding site that according to the experiments is not occupied by our molecules.
- instead, we have conducted a covalent docking of the best inhibitor PB-3 in the newly identified binding site in the cholesterol binding domain, which we show in the paper in Figure 4. This shown binding mode was generated by docking and energy minimization and we have added a paragraph on the used methods in the method section. MD simulations were only employed to investigate the flexibility of the empty binding pocket. We have adjusted the text accordingly.
- reviewer and editorial team suggest us to remove all notions on MD. We are happy to follow this suggestion, as these data are not central to the findings of our work and as the binding hypothesis underlying these MD simulations has been disproven by a long list of experimental data as summarized above.

Apart from these major concerns I suggest to correct/improve following minor issues:

Lines 157-158: " PB-3 was the most active PLY inhibitor of 13 structural derivatives, PB-3.1-PB-3.13" here PB-3 equals PB3-1 as it is missing in the Supplementary figure 8, but this is not stated anywhere. Writing something like "PB-3 was significantly more active than any of its 12 derivatives PB-3.2-PB-3.13." would increase the understandability.

No, PB-3.1 is not identical with PB-3. It is the aldehyde precursor of PB-3 as written in the text, shown in Figure 2b,c and in the methods part. To make this clearer we have added an additional hint to the structure in the brackets following "PB-3.1-PB-3.13 (Supplementary Figure 8, Figure 2b,c)".

Figure 2a. I have printed the pdf on our HPC laser printer and can not distinguish the colors for PB-1 and PB-2. using a different color, e.g. for PB-1 would be of a great help.

We have changed Figure 2a now using more distinct colors for the curves of PB-1 and PB-2 which should be better visually distinguishable.

Figure 2. in the legend subfigure g is mentioned, but not subfigure g is present

Thank you very much for this hint. Indeed, subfigure 2g showing the kinetic and binding affinity data obtained from BLI has been cut out in the second revised version. We have added it again and extend the legend explaining its content.

line 266 typo: "inactive hemolysis"

corrected

line 290 corrupted sentence (and suggested a binding mode in which PB-3 through hydrophobic interactions with the amino acid residues of domain 4 with (Figure 4f))."

Thanks for the hint, corrected to: "suggested binding of **PB-3** through hydrophobic interactions and H-bonding with the amino acid residues of domain 4."

lines 306-318: yellow section on perforin doesn't fit to the flow of this paragraph, I would move it to the next paragraph as an additional independent control. or make a new paragraph...

Done, we have made a new paragraph.

line 347 "All three PB-compounds did not inhibit ..." -> "None of the three PB-compounds inhibited..." corrected.

lines 349-357 speak mostly about PB-3, therefore I would move the statement "The protective effect of PB-2" to the end of the paragraph. PB-3... similar protective effects were also observed for PB-2 (supplementary Figure 9). otherwise the discussion of the results for PB-2 is missing in the paragraph.

We have changed this sentence to: "The protective effect of PB-3 against PLY in A549 cells was illustrated by using confocal laser scanning microscopy (Figure 5, similar effects of PB-2 are shown in Supplementary Figure 9).

line 363. "nuclear executioner caspase activation" what is this? executioner of caspase activation?

Caspases 3,6, and 7 are denominated as "nuclear executioner caspases" being responsible for the execution of apoptosis. For simplicity we have modified the statement to "caspase activation".

The legend of Figure 4 is corrupt in multiple ways. line 375 typo "muatnt", line 378 "and unable to neutralize" -> "and is unable to neutralize", same line typo "concntrations" line 382 "PLY (e) intermedilysin" -> "PLY. (e) Intermedilysin"; the description of the distinct samples in f belongs to e. Legend f: " A docking suggested binding mode of PB-3." -> " A binding mode of PB-3 suggested by docking..." some more details??? you described nowhere the docking of PB-3

All corrected. The covalent docking of PB-3 has been described in an additional subsection of the methods part entitled "Method of covalent docking".

Discussion. The sentence " Binding of PB-2 inhibited membrane insertion of PLY and the formation of PLY pores as visualized by cryo-transmission electron tomography of cholesterol-containing liposomes and resulted in protein precipitation." is not connected to the sentence before and afterwards as it is not clear why cryo-TEM was performed for PB-2 only?

For a better connection, we begin the sentence now with "PB-2 inhibited"

As explained above, at the time of these experiments, PB-3 was not yet available. All experimental data collected on PB-3 indicate, however, that it is an even more potent inhibitor of pore formation than PB-2.

Methods: details on MD simulations like force field, protein structure (monomer? dimer?) presence of lipids? ion concentration? and details on analysis missing.

Details on MD simulations have been deleted. Details on virtual screening have been described in the manuscript text and in the methods section including the starting protein structures used, the generation of the dimer structure, and the forcefield used (MMFF94). A new paragraph on the covalent docking method has been added.

With this, we hope to have answered your questions regarding our manuscript comprehensively and to your satisfaction,

yours sincerely,

Jörg Rademann

REVIEWERS' COMMENTS

Reviewer #4 (Remarks to the Author)

I thank the authors for careful considerations of all my questions.

I would have loved to see MD simulation of all three inhibitors (PB-1 to PB-3) in the "correct" binding pocket, which was identified by mutagenesis later in the manuscript and I think that they could deliver really interesting insights. However, I can fully understand that it is too late during the revision process of the manuscript to generate these data.

The authors answered all my major concerns, thus I have only a few minor correction-suggestions that could increase the clarity of the manuscript for a broad audience.

Results: first paragraph. you speak about two potential binding sites. in line 92 you speak about "the proposed binding site"- which one is it? the one in the cholesterol binding domain or in the oligomerization domain?

Figure legend 1c, please add the info that the oligomerization domain site was targeted in the docking

line 135: what does SAR stand for please?

line 157. if PB-3 is not equal to PB-3.1 then the following sentence "PB-3 was the most active PLY inhibitor of 13 structural derivatives, PB-3.1-PB-3.13 " should call something like PB-3 was more active than any of its 13 structural derivatives, PB-3.1-PB-3.13...

Figure 4f: "Covalent binding mode of PB-3 suggested docking, " -> Binding mode of PB-3 suggested by covalent docking,

I congratulate on and thank you for your stamina and scientific curiosity concerning this work. it must have been a long and complicated journey, but the results and the manuscript definitely pay off!